# LEARNING-AUGMENTED STREAMING ALGORITHMS FOR CORRELATION CLUSTERING

## ABSTRACT

We study streaming algorithms for Correlation Clustering. Given a complete graph as an arbitrary-order stream of edges, with each edge labelled as positive or negative, the goal is to partition the vertices into disjoint clusters, such that the number of disagreements is minimized. In this paper, we give the first learning-augmented streaming algorithms for the problem, achieving the first better-than-3-approximation in dynamic streams. Our algorithms draw inspiration from recent works of Cambus et al. (SODA'24), and Chakrabarty and Makarychev (NeurIPS'23). Our algorithms use the predictions of pairwise dissimilarities between vertices provided by a predictor and achieve an approximation ratio that is close to $2.06$ under good prediction quality. Even if the prediction quality is poor, our algorithms cannot perform worse than the well known PIVOT algorithm, which achieves a 3-approximation. Our algorithms are much simpler than the recent $1.847$-approximation streaming algorithm by Cohen-Addad et al. (STOC'24) which appears to be challenging to implement and is restricted to insertion-only streams. Experimental results on synthetic and real-world datasets demonstrate the superiority of our proposed algorithms over their non-learning counterparts.

## 1 INTRODUCTION

Correlation Clustering is a fundamental problem in machine learning and data mining, and it has a wide range of applications, such as image segmentation (Kim et al., 2014), community detection (Shi et al., 2021), automated labeling (Chakrabarti et al., 2008), etc. Given a complete graph $G = (V, E = E^+ \cup E^-)$, where each edge is labeled as positive $(+)$ or negative $(-)$, the goal is to find a clustering $\mathcal{C}$, i.e., a partition of $V$ into disjoint clusters $C_1, C_2, \ldots, C_t$, where $t$ is arbitrary, that minimizes the following cost:

$$\text{cost}_G(\mathcal{C}) := |\{(u, v) \in E^+ : \exists i \neq j : u \in C_i, v \in C_j\}| + |\{(u, v) \in E^- : \exists i : u, v \in C_i\}|.$$

That is, the number of negative edges in the same cluster plus the number of positive edges between different clusters. (We often refer to as the number of *disagreements*.)

This problem, introduced by Bansal et al. (2004), is known to be APX-hard (Charikar et al., 2005). Hence, significant efforts have been dedicated to designing approximation algorithms for this problem (Bansal et al., 2004; Charikar et al., 2005; Ailon et al., 2008; Chawla et al., 2015; Cohen-Addad et al., 2022; 2023; Cao et al., 2024; Cohen-Addad et al., 2024b), culminating in a $1.437$-approximation via a linear program (LP) based rounding (Cao et al., 2024). There exists a purely combinatorial algorithm that achieves a $(2 - 2/13 + \varepsilon)$-approximation (Cohen-Addad et al., 2024b).

Partially due to storage limitations and the rapidly growing volume of data, graph streaming algorithms for Correlation Clustering have received increasing attention recently. In this setting, a graph is represented as a sequence of edge insertions or deletions, known as a *graph stream*. The objective is to scan the sequence in a few number of passes, ideally, 1 pass and find a high-quality clustering of the vertex set with a low Correlation Clustering cost, while minimizing space usage. If the sequence contains only edge insertions, it is referred to as an *insertion-only* stream; if both insertions and deletions are allowed, it is referred to as a *dynamic* stream. Since the output of the clustering inherently requires $\Omega(n)$ bits of space (as each vertex needs a label to indicate its cluster membership), most previous research has primarily focused on the *semi-streaming* model, i.e., the algorithm

is allowed to use $\widetilde{O}(n) := O(n \operatorname{polylog} n)$ space[1]. Actually, there exists a single-pass $(1 + \varepsilon)$-approximation algorithm in the semi-streaming model even for dynamic streams (Ahn et al., 2021; Behnezhad et al., 2023). However, this algorithm takes exponential time, as it enumerates all possible clusterings, evaluates their costs, and outputs the minimum using the cut sparsifier. Therefore, previous works have focused on designing polynomial-time algorithms (Cohen-Addad et al., 2021; Assadi & Wang, 2022; Behnezhad et al., 2022; 2023; Chakrabarty & Makarychev, 2023; Cambus et al., 2024). Notably, Chakrabarty & Makarychev (2023) and Cambus et al. (2024) independently proposed single-pass $(3 + \varepsilon)$-approximation algorithms recently. The former is only applicable to insertion-only streams, whereas the latter works in the dynamic setting.

For a long time, achieving a 3-approximation has been considered a natural target in the streaming setting, while recently Cohen-Addad et al. (2024b) gave a $(2 - 2/13 + \varepsilon)$-approximation for this problem under insertion-only streams in $O(2^{\varepsilon^{-O(1)}} n \log n)$ space. Though beautiful in theory, their algorithm (and even its other variants that achieve better than 3 approximation) is based on local search, while in turn requires to enumerate a large number of subsets of a constant-size set $S$. Such an enumeration is considered to be quite impractical, as $|S|$ is a very large constant. On the other hand, all the previous $(3 + \varepsilon)$-approximation algorithms in the streaming model are quite simple and much easier for implementation. Therefore, a natural question arises:

> *Is it possible to obtain a **practical**, better-than-3-approximation algorithm for Correlation Clustering in both insertion-only and dynamic streams?*

We affirmatively answer the above question by leveraging ideas from *learning-augmented algorithms (LAAs)*. An LAA uses *predictions* to enhance its performance. These algorithms stem from practical scenarios where machine learning techniques exploit data structure to exceed the worst-case guarantees of traditional algorithms. Our LAAs fit into the category of learning-augmented streaming algorithms (Hsu et al., 2019; Jiang et al., 2020; Chen et al., 2022a; Aamand et al., 2023). It is worth mentioning that both our work and previous efforts on learning-augmented streaming algorithms mainly focus on using predictors to improve the corresponding space-accuracy tradeoffs.

Now, we describe the prediction we are considering. We assume that the algorithm has oracle access to a predictor $\Pi : \binom{V}{2} \to [0, 1]$ that predicts the *pairwise dissimilarities*[2] $d_{uv}$ between any two vertices $u$ and $v$ in $V$. We believe such predictors are natural and arise in many situations. Indeed, it is quite common that *multiple* graphs are defined over the same set of vertices. Patients in a healthcare system can be represented by vertices, and multiple networks can be defined based on different types of relationships, such as shared medical conditions (disease networks), visits to the same healthcare providers (provider networks), or being part of the same clinical trials. In a biological context, the vertex set could represent genes or proteins. One network might capture protein-protein interactions, while another could represent gene co-expression levels. Additionally, metabolic or signaling pathways might define other networks. It is possible to leverage machine learning or data mining techniques to learn the pairwise (dis)similarities between nodes using one or more of these networks. If two patients (or genes/proteins) are found to be similar in one network, it is quite possible they will exhibit similar behavior in other networks as well. *Leveraging these similarities across networks can greatly aid in exploring the cluster structure of any newly defined network over the same set of vertices.* A similar situation arises with temporal graphs, where a sequence of graphs over the same set of vertices has different edge sets across different time slots. Useful information, such as vertex pairwise (dis)similarities learned in the past, can be exploited to extract structural insights from the graph in the present or future time frames. Finally, we remark that several other works have considered similar oracles for pairwise (dis)similarity in different contexts (e.g., in the query model (Silwal et al., 2023; Kuroki et al., 2024)).

By using the above predictions, we give the first LAA for Correlation Clustering that beats 3-approximation if the predictions are good, while still achieves $(3 + \varepsilon)$-approximation even if the predictor behaves poorly. That is, our algorithm is both robust and consistent, as desired for most natural LAAs (Mitzenmacher & Vassilvitskii, 2021). Furthermore, our algorithms are simple and easily implementable. We will use a parameter $\beta \in [1, \infty)$ to measure the quality of our predictor. Informally, we call a predictor $\beta$-level if the cost of the predictions induced clustering is at most a

---

[1]On the other hand, Assadi et al. (2023) studied streaming algorithms using $\operatorname{polylog} n$ bits of space for estimating the optimum Correlation Clustering *cost*, while their algorithms do not find the clustering.

[2]Note that one can directly treat $1 - d_{uv}$ as the pairwise *similarity* between $u, v$.

Table 1: Comparison of our results with the best-known space-approximation tradeoffs. Here, $\varepsilon \in (0, 1)$ and $\beta \geq 1$. All space complexities are measured in words. All algorithms use a single pass.

| Streaming Model | Best Space & Approximation Tradeoffs (Without Predictions) | Our Results |
|---|---|---|
| dynamic | $(3 + \varepsilon)$-approx., $O(\varepsilon^{-1} n \log^4 n)$ space (Cambus et al., 2024) | $(\min\{2.06\beta, 3\} + \varepsilon)$-approx., $O(n \log^6 n + \varepsilon^{-2} n \log^5 n)$ space |
| insertion-only | $(2 - 2/13 + \varepsilon)$-approx., $O(2^{\varepsilon^{-O(1)}} n \log n)$ space (Cohen-Addad et al., 2024b) | $(\min\{2.06\beta, 3\} + \varepsilon)$-approx., $O(\varepsilon^{-2} n \log n)$ space |

$\beta$ factor of the cost of the optimal solution. (We refer to Definition 2.1 for the formal definition of a $\beta$-level predictor.) That is, the smaller $\beta$ is, the higher the quality of the predictor. Our results are summarized in Table 1. Specifically, for dynamic streams, we have the following theorem. (In the following, "with high probability" refers to the probability of at least $1 - 1/n^c$ for some constant $c > 0$.)

**Theorem 1.1.** *Let $\varepsilon \in (0, 1/4)$ and $\beta \geq 1$. Given oracle access to a $\beta$-level predictor, there exists a single-pass streaming algorithm that provides an expected $(\min\{2.06\beta, 3\} + \varepsilon)$-approximation for Correlation Clustering in dynamic streams with high probability. The algorithm uses $O(n \log^6 n + \varepsilon^{-2} n \log^5 n)$ words of space.*

Note that our algorithm achieves a better-than-3 approximation in dynamic streams under good prediction quality, while the previous best-known algorithm in dynamic streams is a $(3 + \varepsilon)$-approximation due to Cambus et al. (2024).

Furthermore, we also obtain an algorithm in insertion-only streams, which is different from the algorithm in dynamic streams while achieving the same approximation guarantee with improved space complexity.

**Theorem 1.2.** *Let $\varepsilon \in (0, 1)$ and $\beta \geq 1$. Given oracle access to a $\beta$-level predictor, there exists a single-pass streaming algorithm that provides an expected $(\min\{2.06\beta, 3\} + \varepsilon)$-approximation for Correlation Clustering in insertion-only streams with high probability. The algorithm uses $O(\varepsilon^{-2} n \log n)$ words of space.*

Note that it is standard to assume that the space of the oracle is not included in the space usage of our algorithms, as is common in learning-augmented streaming algorithms (Hsu et al., 2019; Jiang et al., 2020; Chen et al., 2022a; Aamand et al., 2023). As noted in (Hsu et al., 2019), reliable predictors can often be learned in a space-efficient manner in practice. Furthermore, as stated before, to cluster a graph, we may use ML methods to train some other related networks that are defined on the same vertex set, to learn the pairwise (dis)similarities. In particular, we can learn the node embeddings from these related networks, which map all vertices to Euclidean space. Then the distances between these points serve naturally as pairwise dissimilarities and satisfy the triangle inequality.

To complement our theoretical results, we conduct comprehensive experiments to evaluate our algorithms on both synthetic and real-world datasets. Experimental results demonstrate the superiority of our LAAs.

### 1.1 TECHNICAL OVERVIEW

Our LAAs rely on the influential PIVOT algorithm by Ailon et al. (2008) and the LP rounding algorithm by Chawla et al. (2015). The PIVOT algorithm begins by selecting a random permutation $\pi$ over the vertices of the graph. It then iteratively forms clusters by choosing the vertex with the smallest rank according to $\pi$, along with its neighbors in the graph. Once a cluster is formed, it is removed from the graph. This process continues until all vertices have been assigned to clusters. The LP rounding algorithm first solves an LP corresponding to Correlation Clustering, and then applies a PIVOT-based algorithm using the LP solution. Next, we describe our algorithms. The high-level idea is to incorporate the above LP rounding approach with the "truncated" PIVOT algorithms (Cambus et al., 2024; Chakrabarty & Makarychev, 2023), where our predictions correspond to a feasible LP solution in some sense. Specifically, for dynamic streams, we maintain a certain number of $\ell_0$-samplers during the stream and derive a truncated subgraph at the end of the stream. Then we run

the PIVOT algorithm and the LP rounding style algorithm on the subgraph respectively and obtain two clusterings. Finally, we output the clustering with the lower cost. For insertion-only streams, we employ two different methods respectively to store at most $k$ neighbors for each vertex during the stream. The first method is similar to the PIVOT algorithm and the second method is similar to the LP rounding style algorithm. Then we run the PIVOT algorithm on two stored subgraphs and output the clustering with the lower cost. We note that the cost of a clustering cannot be exactly calculated during the stream, since our algorithms cannot store the entire graph. Therefore, we apply the graph sparsification techniques to approximate the clustering cost within a multiplicative factor of $(1 \pm \varepsilon)$.

The analysis is non-trivial, even in insertion-only streams. We categorize all clusters into pivot clusters and singleton clusters, and analyze their costs respectively. Our key observation is that the truncated version of the LP rounding algorithm is equivalent to the algorithm that first samples a subgraph $G'$ according to the predictions and then runs the "truncated" PIVOT algorithms on $G'$. Our main technical contribution is to prove that 1) the cost of pivot clusters produced by the truncated version of the LP rounding algorithm is at most $2.06\beta$ times the cost of optimal solution (Lemma 3.6 and Lemma 4.2); 2) the optimal solution on $G'$ does not differ from the optimal solution on the original graph $G$ by a lot (Lemma 4.3). In this way, our algorithms can keep the space small while achieving an approximation ratio better than 3 under good prediction quality.

## 2 PRELIMINARIES

**Notations.** Throughout the paper, we let $G = (V, E)$ be an undirected and unweighted complete graph with $|V| = n$, $|E| = m$, where each edge is labeled as positive or negative (i.e., $E = E^+ \cup E^-$). In some places of the paper, we identify the input graph only with the set of positive edges, i.e., $G^+ = (V, E^+)$ and the negative edges are defined implicitly. For each vertex $u \in V$, let $N(u)$ be the set of all neighbors of $u$ and $N^+(u)$ be the set of positive neighbors of $u$ (i.e., vertices that are connected by a positive edge). Correspondingly, let $\deg(u) := |N(u)|$ be the degree of $u$, and similarly, $\deg^+(u) := |N^+(u)|$. We use $\text{cost}_G(\mathcal{C})$ to denote the cost of the clustering $\mathcal{C}$ on $G$. We say an algorithm achieves an $\alpha$-approximation if it outputs a clustering $\mathcal{C}$ on $G$ such that $\text{OPT} \leq \text{cost}_G(\mathcal{C}) \leq \alpha \cdot \text{OPT}$, where OPT denotes the cost of an optimal solution on $G$.

Next, we give the formal definition of a $\beta$-level predictor.

**Definition 2.1** ($\beta$-level predictor). For any $\beta \geq 1$, we call a predictor $\beta$-*level*, if it predicts the pairwise dissimilarities $d_{uv}$ between any two vertices $u$ and $v$ in $V$ such that (1) (triangle inequality) $d_{uv} + d_{vw} \geq d_{uw}$ for all $u, v, w \in V$, (2) $\sum_{(u,v) \in E^+} d_{uv} + \sum_{(u,v) \in E^-} (1 - d_{uv}) \leq \beta \cdot \text{OPT}$.

Intuitively, a smaller $\beta$ indicates a higher-quality predictor, and in this case $d_{uv}$ can be used to determine how likely $u$ and $v$ are in the same cluster of the optimal solution. However, we point out that the predictions can be completely independent of the input graph. In the worst case, the predictions can be arbitrary, which is allowed for LAAs since robustness is a desired goal.

We remark that both our definition of the $\beta$-level predictor and our algorithms are inspired by an approximation algorithm given by Chawla et al. (2015), who give an LP based algorithm for Correlation Clustering achieving 2.06-approximation and in some sense, the $\beta$-level predictor corresponds to a solution to the LP in Chawla et al. (2015).

Due to space limitations, we introduce the useful tools utilized in this paper in Appendix B.

## 3 OUR ALGORITHM IN DYNAMIC STREAMS

### 3.1 OFFLINE IMPLEMENTATION

**Overview.** For ease of illustration of ideas, we first describe our algorithm in the offline setting. The overall framework is similar to (Cambus et al., 2024). The algorithm takes $G^+ = (V, E^+)$ as input. Initially, we pick a random permutation $\pi$ over the set of vertices. Then we divide all vertices into interesting and uninteresting vertices based on the relationship between the rank and the positive degree of a vertex. Specifically, a vertex $u$ is uninteresting if $\pi_u \geq \tau_u$ where $\tau_u := \frac{c}{\varepsilon} \cdot \frac{n \log n}{\deg^+(u)}$ (or equivalently $\deg^+(u) \geq \sigma_u$ where $\sigma_u := \frac{c}{\varepsilon} \cdot \frac{n \log n}{\pi_u}$), and interesting otherwise. Here, $\varepsilon \in (0, 1/4)$ and $c$ is a universal large constant. Finally we run two pivot-based algorithms on the subgraph $G_{\text{store}}$ induced by the set of interesting vertices and output the clustering with the lower cost. We defer its pseudocode (Algorithm 2) to Appendix C.

Note that in the clustering phase, we apply two pivot-based approaches on the truncated graph $G_{\text{store}}$: Algorithms TRUNCATEDPIVOT and TRUNCATEDPIVOTWITHPRED.

**Algorithm TRUNCATEDPIVOT.** This algorithm simulates the Parallel Truncated-Pivot algorithm by Cambus et al. (2024) and produces the same clustering. This algorithm proceeds in iterations. Let $U^{(t)}$ denote the set of unclustered vertices in $G_{\text{store}}$ at the beginning of iteration $t$. Initially, all the interesting vertices are unclustered. At the beginning of iteration $t$, if $U^{(t)} \neq \emptyset$, then we pick the vertex $u$ from $U^{(t)}$ with the smallest rank. Then we mark it as a pivot and create a pivot cluster $S^{(t)}$ containing $u$ and all of its unclustered positive neighbors in $G_{\text{store}}$. At the end of iteration $t$, we remove all vertices clustered in this iteration from $U^{(t)}$. Then the algorithm proceeds to the next iteration. If $U^{(t)} = \emptyset$ at the beginning of iteration $t$, then we know that all the interesting vertices are clustered. Now it suffices to assign each uninteresting vertex to a cluster. Each uninteresting vertex $u$ joins the cluster of pivot $v$ with the smallest rank if $(u, v) \in E^+$ and $\pi_v < \tau_u$. Then each unclustered vertex $u \in V$ creates a singleton cluster. Finally, we output all pivot clusters and singleton clusters. We defer its pseudocode (Algorithm 3) to Appendix C.

**Algorithm TRUNCATEDPIVOTWITHPRED.** This algorithm has oracle access to a $\beta$-level predictor $\Pi$. The algorithm closely resembles Algorithm TRUNCATEDPIVOT. The differences are as follows: (1) At iteration $t$, we create a pivot cluster $S^{(t)}$ containing $u$ and add all the unclustered vertices $v$ in $G_{\text{store}}$ to $S^{(t)}$ with probability $(1 - p_{uv})$ independently, where $p_{uv} = f(d_{uv})$ and $d_{uv} = \Pi(u, v)$. If $(u, v) \in E^+$, then $f(d_{uv}) = f^+(d_{uv})$; otherwise $f(d_{uv}) = f^-(d_{uv})$. We set $f^+(x)$ to be 0 if $x < a$, $\left(\frac{x-a}{b-a}\right)^2$ if $x \in [a, b]$, and 1 if $x > b$, where $a = 0.19$ and $b = 0.5095$; we set $f^-(x) = x$. (2) Each uninteresting vertex $u$ joins the cluster of pivot $v$ in the order of $\pi$ with probability $(1 - p_{uv})$ independently, if $\pi_v < \tau_u$. We defer its pseudocode (Algorithm 4) to Appendix C.

In Section 3.3, we will prove the following theorem that gives a theoretical guarantee of the offline algorithm.

**Theorem 3.1.** *Let $\varepsilon \in (0, 1/4)$ and $\beta \geq 1$. Given oracle access to a $\beta$-level predictor, Algorithm 2 provides an expected $(\min\{2.06\beta, 3\} + \varepsilon)$-approximation for Correlation Clustering.*

### 3.2 IMPLEMENTATION IN DYNAMIC STREAMS

In this subsection, we implement the offline algorithm in dynamic streams, as shown in Algorithm 1. A key observation is that it suffices to store the positive edges incident to interesting vertices since we apply pivot-based algorithms on the subgraph induced by interesting vertices and then try to assign uninteresting vertices to pivot clusters. To this end, we maintain a certain number of $\ell_0$-samplers for each vertex, which can be achieved in the dynamic semi-streaming model (Jowhari et al., 2011). As we will see in the analysis, the $\ell_0$-samplers allow us to recover the edges incident to all the interesting vertices with high probability. Thus we can simulate the clustering phase of the offline algorithm. Specifically, we simulate Algorithms TRUNCATEDPIVOT and TRUNCATEDPIVOTWITHPRED using the stored information, and output the clustering with the lower cost.

Note that in the final step, the cost of a clustering cannot be exactly calculated, as our streaming algorithm cannot store the entire graph. To overcome this challenge, we borrow the idea from (Behnezhad et al., 2023) and utilize the graph sparsification technique (Ahn et al., 2012) to estimate the cost. Specifically, during the streaming phase, we maintain a cut sparsifier $H^+$ for the subgraph $G^+$. Let AGM-SPARSIFICATION be any algorithm for constructing a cut sparsifier that satisfies the guarantee in (Ahn et al., 2012) (see Appendix B). For each item $s_i = (e_i = (u, v), \Delta_i)$ in the dynamic stream, where $\Delta_i \in \{-1, 1\}$ indicates the insertion or deletion of $e_i$, we apply AGM-SPARSIFICATION($H^+, s_i$) to determine whether $(u, v)$ belongs to $H^+$ and, if so, its corresponding weight in $H^+$. We also maintain the positive degree $\deg^+(u)$ of each vertex $u$. Then we can approximate the cost of a clustering up to a $(1 \pm \varepsilon)$-multiplicative error with high probability.

### 3.3 PROOF OF THEOREM 3.1

As the final clustering produced by the offline algorithm is the lower-cost one produced by two pivot-based algorithms, we start by analyzing the costs of these two clusterings (i.e., Lines 5 and 6 of Algorithm 2). For ease of analysis, we separately examine the approximation ratios of the equivalent versions (Algorithms CKLPU-PIVOT and PAIRWISEDISS) that produce these two clusterings.

**Algorithm CKLPU-PIVOT** (Algorithm 4 in (Cambus et al., 2024)). This algorithm proceeds in iterations. Let $U^{(t)}$ denote the set of unclustered vertices at the beginning of iteration $t$. Initially, we

---

**Algorithm 1** A dynamic streaming algorithm for Correlation Clustering with predictions

---

**Input:** Graph $G^+ = (V, E^+)$ as an arbitrary-order dynamic stream of edges, oracle access to a $\beta$-level predictor $\Pi$

**Output:** Partition of $V$ into disjoint sets

    ▷ **Preprocessing phase**

1: Pick a random permutation of vertices $\pi : V \to \{1, \ldots, n\}$.
2: **for** each vertex $u \in V$ **do**
3:      Let $\deg^+(u) \leftarrow 0$. Mark $u$ as unclustered and interesting.
4:      Let $\sigma_u := \frac{c}{\varepsilon} \cdot \frac{n \log n}{\pi_u}$, where $c$ is a universal large constant.
5:      Initialize $10c \log n \cdot \sigma_u$ independent $\ell_0$-samplers (with failure probability $1/10$) for the adjacency vector of $u$ (the row of the adjacency matrix of $G^+$ that corresponds to $u$).
6: Initialize a cut sparsifier $H^+$ for $G^+$.

    ▷ **Streaming phase**

7: **for** each item $s_i = (e_i = (u, v), \Delta_i)$ in the dynamic stream **do**
8:      Update $\deg^+(u)$, $\deg^+(v)$ and all the $\ell_0$-samplers associated with $u$ and $v$.
9:      Apply AGM-SPARSIFICATION$(H^+, s_i)$.

    ▷ **Postprocessing phase**

10: A vertex $u$ marks itself uninteresting if $\deg^+(u) \geq \sigma_u$.
11: Retrieve all incident edges of interesting vertices (with high probability) using the $\ell_0$ samplers.
12: Let $G_{\text{store}}$ be the graph induced by the interesting vertices.
13: $\mathcal{C}_1 \leftarrow \text{TRUNCATEDPIVOT}(G^+, G_{\text{store}}, \pi)$
14: $\mathcal{C}_2 \leftarrow \text{TRUNCATEDPIVOTWITHPRED}(G^+, G_{\text{store}}, \pi, \Pi)$
15: $\widetilde{\text{cost}}_G(\mathcal{C}_1) \leftarrow \sum_{C \in \mathcal{C}_1}(\frac{1}{2}\delta_{H^+}(C) + \binom{|C|}{2} - \frac{1}{2}\sum_{u \in C}\deg^+(u))$
16: $\widetilde{\text{cost}}_G(\mathcal{C}_2) \leftarrow \sum_{C \in \mathcal{C}_2}(\frac{1}{2}\delta_{H^+}(C) + \binom{|C|}{2} - \frac{1}{2}\sum_{u \in C}\deg^+(u))$
17: $i \leftarrow \arg\min_{i=1,2}\{\widetilde{\text{cost}}_G(\mathcal{C}_i)\}$.
18: **return** $\mathcal{C}_i$

---

pick a random permutation $\pi$ over vertices, and all the vertices are unclustered. At the beginning of iteration $t$, let $\ell_t = \frac{c}{\varepsilon} \cdot \frac{n \log n}{t}$. Each unclustered vertex $v$ with $\deg^+(v) \geq \ell_t$ creates a singleton cluster. We pick the $t$-th vertex $u$ in $\pi$. If $u$ is unclustered, then we mark it as a pivot and create a pivot cluster $S^{(t)}$ containing $u$ and all of its unclustered positive neighbors. At the end of iteration $t$, we remove all vertices clustered in this iteration from $U^{(t)}$. Then the algorithm proceeds to the next iteration. Finally, we output all pivot clusters and singleton clusters. We defer its pseudocode (Algorithm 5) to Appendix C.

**Algorithm PAIRWISEDISS.** This algorithm has oracle access to a $\beta$-level predictor $\Pi$. The only difference from Algorithm CKLPU-PIVOT is that at iteration $t$, we create a pivot cluster $S^{(t)}$ containing $u$ and add all unclustered vertices $v$ to $S^{(t)}$ with probability $(1 - p_{uv})$ independently, where $p_{uv} = f(d_{uv})$ and $d_{uv} = \Pi(u, v)$. We defer its pseudocode (Algorithm 6) to Appendix C.

### 3.3.1 THE OFFLINE ALGORITHM AS A COMBINATION OF CKLPU-PIVOT AND PAIRWISEDISS

We first show that if the offline algorithm (Algorithm 2) and Algorithm CKLPU-PIVOT (resp. PAIRWISEDISS) use the same randomness, then Algorithm CKLPU-PIVOT (resp. PAIRWISEDISS) and Line 5 (resp. Line 6) of Algorithm 2 output the same clustering.

**Lemma 3.2** (Lemma 8 in Cambus et al. (2024)). *If Algorithm 2 and Algorithm* CKLPU-PIVOT *use the same permutation $\pi$, then Algorithm* CKLPU-PIVOT *and Line 5 of Algorithm 2 output the same clustering of $V$.*

**Lemma 3.3.** *If Algorithm 2 and Algorithm* PAIRWISEDISS *use the same permutation $\pi$ and predictions $\{d_{uv}\}_{u,v \in V}$, then Algorithm* PAIRWISEDISS *and Line 6 of Algorithm 2 output the same clustering of $V$ with the same probability.*

### 3.3.2 THE APPROXIMATION RATIOS OF CKLPU-PIVOT AND PAIRWISEDISS

Now it suffices to analyze Algorithm CKLPU-PIVOT and Algorithm PAIRWISEDISS respectively. We follow the analysis framework in (Cambus et al., 2024). Specifically, we analyze the cost of pivot clusters and singleton clusters, respectively. For the former, we can directly apply the analysis

of original pivot-based algorithms (Ailon et al., 2008; Chawla et al., 2015), where we only focus on a subset of vertices (i.e., $V \setminus V_{\text{sin}}$ where $V_{\text{sin}}$ is the set of singletons). For the latter, we divide all the positive edges incident to singleton clusters (denoted as $E_{\text{sin}}$) into good edges (denoted as $E_{\text{good}}$) and bad edges (denoted as $E_{\text{bad}}$). Specifically, we define an positive edge incident to a singleton cluster to be good if the other endpoint was included in a pivot cluster *before* the singleton was created. Otherwise, the edge is bad. In other words, bad edges are those that either connect two singletons or the other endpoint was included in a pivot cluster *after* the singleton was created.

In this way, we can charge the cost of good edges to the cost of pivot clusters. Therefore, it suffices to bound the cost of bad edges. The following lemma shows that we can relate the cost of bad edges to the cost of pivot clusters, and thus bound the cost of the final clustering.

**Lemma 3.4** (Cambus et al. (2024)). *Let $\varepsilon \in (0, 1/4)$. Let $P$ denote the cost of pivot clusters, and let $W$ denote the cost of the clustering returned by the algorithm, then $\mathbb{E}[W] = \mathbb{E}[P + |E_{\text{bad}}|] \leq (1 + 4\varepsilon) \cdot \mathbb{E}[P] + \frac{1+4\varepsilon}{n^{\alpha-2}}$, where $\alpha := c/2 - 1 \gg 2$.*

Now we are ready to analyze the approximation ratios of Algorithms CKLPU-PIVOT and PAIR-WISEDISS. We have the following lemma, which states the approximation guarantee of Algorithm CKLPU-PIVOT, and thus that of the clustering returned by Line 5 of Algorithm 2.

**Lemma 3.5** (Cambus et al. (2024)). *Let $\varepsilon \in (0, 1/4)$. Let $\mathcal{C}_1$ denote the clustering returned by Line 5 of Algorithm 2, then $\mathbb{E}[\text{cost}_G(\mathcal{C}_1)] \leq (3 + 12\varepsilon) \cdot \text{OPT} + \frac{1+4\varepsilon}{n^{\alpha-2}}$, where $\alpha := c/2 - 1 \gg 2$.*

Next, we focus on the analysis of Algorithm PAIRWISEDISS.

**Lemma 3.6.** *Let $P_2$ denote the cost of pivot clusters returned by Algorithm PAIRWISEDISS. We have $\mathbb{E}[P_2] \leq 2.06\beta \cdot \text{OPT}$.*

*Proof.* Consider iteration $t$ of Algorithm PAIRWISEDISS, if vertex $u$ considered in this iteration is unclustered (i.e., $u \in U^{(t)}$), then we call iteration $t$ a pivot iteration. The key observation is that the pivot iterations in Algorithm PAIRWISEDISS are equivalent to the iterations of 2.06-approximation LP rounding algorithm by Chawla et al. (2015): given that $u$ is unclustered (i.e., $u \in U^{(t)}$), the conditional distribution of $u$ is uniformly distributed in $U^{(t)}$, and the cluster created during this iteration contains $u$ and all the unclustered vertices $v$ added with probability $(1 - p_{uv})$. Therefore, we can directly apply the triangle-based analysis in (Chawla et al., 2015). Define $L := \sum_{(u,v)\in E^+} d_{uv} + \sum_{(u,v)\in E^-} (1 - d_{uv})$. Since the predictor is $\beta$-level, by Definition 2.1, we have that the predictions $\{d_{uv}\}_{u,v\in V}$ satisfy triangle inequality and $L \leq \beta \cdot \text{OPT}$. It follows that for all pivot iterations $t$, $\mathbb{E}[P_2^{(t)}] \leq 2.06 \cdot \mathbb{E}[L^{(t)}]$, where $P_2^{(t)}$ is the cost induced by the pivot cluster created at iteration $t$, and $L^{(t)} := \sum_{(u,v)\in E^+ \cap E^{(t)}} d_{uv} + \sum_{(u,v)\in E^- \cap E^{(t)}} (1 - d_{uv})$ where $E^{(t)}$ is the set of edges decided at iteration $t$. By linearity of expectation, we have $\mathbb{E}[P_2] = \mathbb{E}[\sum_{t \text{ is a pivot iteration}} P_2^{(t)}] = \sum_{t \text{ is a pivot iteration}} \mathbb{E}[P_2^{(t)}] \leq 2.06 \cdot L \leq 2.06\beta \cdot \text{OPT}$. □

**Corollary 3.7.** *Let $\varepsilon \in (0, 1/4)$. Let $\mathcal{C}_2$ denote the clustering returned by Line 6 of Algorithm 2. We have $\mathbb{E}[\text{cost}_G(\mathcal{C}_2)] \leq (2.06\beta + 8.24\beta\varepsilon) \cdot \text{OPT} + \frac{1+4\varepsilon}{n^{\alpha-2}}$, where $\alpha := c/2 - 1 \gg 2$.*

*Proof of Theorem 3.1.* Theorem 3.1 follows from Lemma 3.5, Corollary 3.7 and Lemma D.3. Note that in Lemma 3.5, we can substitute $\varepsilon' := 12\varepsilon$, where $\varepsilon$ can be arbitrarily small. If $\text{OPT} \geq 1$, then $\mathbb{E}[\text{cost}_G(\mathcal{C}_1)] \leq (3+12\varepsilon) \cdot \text{OPT}$, which gives a $(3+\varepsilon')$-approximation in expectation. If $\text{OPT} = 0$, then $\mathbb{E}[\text{cost}_G(\mathcal{C}_1)] = 1/\text{poly}(n)$. Similarly, in Corollary 3.7, we can substitute $\varepsilon' := 8.24\beta\varepsilon$. □

## 4 AN ALGORITHM IN INSERTION-ONLY STREAMS WITH SMALLER SPACE

**Overview.** We first briefly describe a single-pass $(3 + \varepsilon)$-approximation streaming algorithm by Chakrabarty & Makarychev (2023). Initially, the algorithm adds a positive self-loop for each vertex and picks a random ordering $\pi : V \to \{1, \ldots, n\}$. The rank of $u$ is denoted as $\pi_u$. Then it scans the input stream. For each vertex, the algorithm stores its $k$ positive neighbors with lowest ranks, where $k$ is a constant. Subsequently, it runs the PIVOT algorithm (Ailon et al., 2008) on the stored graph and picks pivots in the order of $\pi$. Finally, it puts unclustered vertices in singleton clusters.

Our main idea is to incorporate the above algorithm with the algorithm from Chawla et al. (2015). Our algorithm uses the predictions of pairwise dissimilarities between any two vertices. We employ two different methods to store at most $k$ neighbors of each vertex. The first method is the same as

Chakrabarty & Makarychev (2023) and the second method is adapted from Chawla et al. (2015), which adds neighbors with probabilities determined by predictions of pairwise dissimilarities. Finally, we obtain two clusterings (denoted as $\mathcal{C}_1$ and $\mathcal{C}_2$) and output the one with the lower cost. Similar to Algorithm 1, here we also need to use the graph sparsification technique (Kelner & Levin, 2011) to approximate the cost of a clustering.

### 4.1 PROOF SKETCH OF THEOREM 1.2

As the final clustering produced by the algorithm is the lower-cost clustering on the two truncated graphs, we start by analyzing the costs of these two clusterings. Similar to the analysis of Algorithm 2, for ease of analysis, we separately examine the approximation ratios of the corresponding offline versions (Algorithms CM-PIVOT and PAIRWISEDISS2) that equivalently output these two clusterings. We defer the proof of equivalence to Appendix F.

**Algorithm CM-PIVOT** (Chakrabarty & Makarychev, 2023)**.** This algorithm proceeds in iterations. Let $F^{(t)}$ denote the set of fresh vertices and $U^{(t)}$ denote the set of unclustered vertices at the beginning of iteration $t$. Additionally, we maintain a counter $K^{(t)}(u)$ for each vertex $u \in V$. Initially, all the vertices are fresh and unclustered, with the counters set to 0. At iteration $t$, we pick a vertex $w^{(t)}$ from the set of fresh vertices $F^{(t)}$ uniformly at random. If $w^{(t)}$ is unclustered, then we mark it as a pivot and create a cluster $S^{(t)}$ containing $w^{(t)}$ and all of its unclustered positive neighbors. Otherwise, we increment the counters for all unclustered positive neighbors of $w^{(t)}$. Subsequently, vertices whose counters reach the value of $k$ are assigned to singleton clusters. At the end of iteration $t$, we remove $w^{(t)}$ from $F^{(t)}$ and remove all vertices clustered in this iteration from $U^{(t)}$. Then the algorithm proceeds to the next iteration. Finally, we output all pivot clusters and singleton clusters. We defer its pseudocode (Algorithm 9) to Appendix E.

**Algorithm PAIRWISEDISS2.** This algorithm has oracle access to a $\beta$-level predictor $\Pi$. This algorithm closely resembles Algorithm CM-PIVOT, differing in the following two aspects: (1) If $w^{(t)} \in U^{(t)}$, then we create a cluster $S^{(t)}$ containing $w^{(t)}$ and add all unclustered vertices $v$ to $S^{(t)}$ with probability $(1 - p_{vw^{(t)}})$ independently, where $p_{vw^{(t)}} = f(d_{vw^{(t)}})$ and $d_{vw^{(t)}} = \Pi(v, w^{(t)})$. (2) If $w^{(t)} \notin U^{(t)}$, we increment the counters for all unclustered vertices $v$ with probability $(1 - p_{vw^{(t)}})$. We defer its pseudocode (Algorithm 10) to Appendix E.

We rely on the analysis framework in Chakrabarty & Makarychev (2023). We categorize all iterations into *pivot iterations* and *singleton iterations*. Both iterations create some clusters. We call the clusters created in pivot iterations *pivot clusters*. Let $P$ denote the cost of all pivot clusters. Therefore, $P = \sum_{t \text{ is a pivot iteration}} P^{(t)}$. Let $S$ denote the cost of all singleton clusters. Therefore, the cost of the algorithm is equal to $P + S$. We have the following guarantee of Algorithm CM-PIVOT.

**Lemma 4.1.** *Let $P_1$ and $S_1$ denote the costs of pivot clusters and singleton clusters, respectively, returned by Algorithm* CM-PIVOT. *Then* $\mathbb{E}[\text{cost}_G(\mathcal{C}_1)] = \mathbb{E}[P_1 + S_1] \leq (3 + \frac{6}{k-1}) \cdot \text{OPT}$.

Next, we analyze Algorithm PAIRWISEDISS2. We first bound the cost of pivot clusters.

**Lemma 4.2.** *Let $P_2$ denote the cost of pivot clusters returned by Algorithm* PAIRWISEDISS2. *We have* $\mathbb{E}[P_2] \leq 2.06\beta \cdot \text{OPT}$.

Next, we bound the cost of singleton clusters returned by Algorithm PAIRWISEDISS2, denoted as $S_2$. We highlight that this part is non-trivial. Different from the analysis in Chakrabarty & Makarychev (2023) which uses a potential function and shows that it is a submartingale, we consider an algorithm equivalent to Algorithm PAIRWISEDISS2. In this algorithm, we construct a random subgraph $G' := (V, E'^+ \cup E'^-)$ where each edge $(u, v) \in E$ is added to $E'^+$ with probability $(1 - p_{uv})$ and added to $E'^-$ with the remaining probability. Then we perform Algorithm CM-PIVOT on $G'$. In other words, we first preround the $\beta$-level predictions $\{d_{uv}\}_{u,v \in V}$ into an new instance $G'$ and then run Algorithm CM-PIVOT on $G'$ where the positive edges are induced by the predictions. We defer its pseudocode (Algorithm 11) to Appendix E.

Therefore, we can apply the guarantee of the cost of singleton clusters returned by Algorithm CM-PIVOT on $G'$. We first show that $G'$ still well preserves the cluster structure of $G$, by showing that the optimal solution on $G'$ does not differ from the optimal solution on $G$ by a lot.

**Lemma 4.3.** $\mathbb{E}[\text{OPT}'] \leq (2\beta + 1) \cdot \text{OPT}$, *where* OPT *is the cost of the optimal solution on $G$ and* OPT' *is the cost of the optimal solution on $G'$.*

*Proof.* Let $\mathcal{C}^*$ be the optimal clustering on $G$ with cost OPT. For any $u, v \in V$, let $x_{uv}^* \in \{0, 1\}$ indicate whether $u$ and $v$ are in the same cluster or not in $\mathcal{C}^*$. Specifically, if $u$ and $v$ are in the same cluster in $\mathcal{C}^*$, then $x_{uv}^* = 0$; otherwise, $x_{uv}^* = 1$. Let $\mathcal{C}'^*$ be the optimal clustering on $G'$ with cost OPT$'$. Then we have

$$\mathbb{E}[\text{OPT}'] = \mathbb{E}[\text{cost}_{G'}(\mathcal{C}'^*)] \leq \mathbb{E}[\text{cost}_{G'}(\mathcal{C}^*)]$$

$$= \sum_{(u,v) \in E^+} [x_{uv}^*(1 - p_{uv}) + (1 - x_{uv}^*)p_{uv}] + \sum_{(u,v) \in E^-} [x_{uv}^*(1 - p_{uv}) + (1 - x_{uv}^*)p_{uv}]$$

$$= \sum_{(u,v) \in E^+} x_{uv}^* + \sum_{(u,v) \in E^-} (1 - x_{uv}^*) + \sum_{(u,v) \in E^+} [p_{uv}(1 - 2x_{uv}^*)] + \sum_{(u,v) \in E^-} [(1 - p_{uv})(2x_{uv}^* - 1)]$$

$$\leq \text{OPT} + \sum_{(u,v) \in E^+} p_{uv} + \sum_{(u,v) \in E^-} (1 - p_{uv})$$

$$\leq \text{OPT} + \sum_{(u,v) \in E^+} 2d_{uv} + \sum_{(u,v) \in E^-} (1 - d_{uv}) \leq (1 + 2\beta) \cdot \text{OPT},$$

where the first step follows from $\text{cost}_{G'}(\mathcal{C}'^*) = \text{OPT}'$, the second step follows from that $\mathcal{C}'^*$ is the optimal clustering on $G'$, the third step follows from our construction of $G'$, the fifth step follows from $\sum_{(u,v) \in E^+} x_{uv}^* + \sum_{(u,v) \in E^-} (1 - x_{uv}^*) = \text{OPT}$ and $\sum_{(u,v) \in E^+} [p_{uv}(1 - 2x_{uv}^*)] + \sum_{(u,v) \in E^-} [(1 - p_{uv})(2x_{uv}^* - 1)] \leq \sum_{(u,v) \in E^+} p_{uv} + \sum_{(u,v) \in E^-} (1 - p_{uv})$ since $1 - 2x_{uv}^* \in \{-1, 1\}$, the sixth step follows from our choice for $p_{uv}$, and the last step follows from $\sum_{(u,v) \in E^+} 2d_{uv} + \sum_{(u,v) \in E^-} (1 - d_{uv}) \leq 2(\sum_{(u,v) \in E^+} d_{uv} + \sum_{(u,v) \in E^-} (1 - d_{uv})) \leq 2\beta \cdot \text{OPT}$. □

Now we are ready to bound the cost of singleton clusters and, consequently, the final clustering returned by Algorithm PAIRWISEDISS2.

**Lemma 4.4.** $\mathbb{E}[S_2] \leq \frac{6(2\beta+1)}{k-1} \cdot \text{OPT}.$

**Corollary 4.5.** $\mathbb{E}[\text{cost}_G(\mathcal{C}_2)] = \mathbb{E}[P_2 + S_2] \leq (2.06\beta + \frac{6(2\beta+1)}{k-1}) \cdot \text{OPT}.$

Therefore, the approximation guarantee of our algorithm in insertion-only streams follows from Lemma 4.1, Corollary 4.5 and Lemma D.3, once we show that the algorithm is an equivalent combination of Algorithms CM-PIVOT and PAIRWISEDISS2.

## 5 EXPERIMENTS

In this section, we evaluate our proposed algorithms empirically on synthetic and real-world datasets. All of our experiments are done on a CPU with i7-13700H processor and 32 GB RAM. All of our algorithms are implemented in Python. For all results, we report the average clustering cost over 20 independent trials. Our source code is available in the supplementary material.

**Datasets. 1) Synthetic datasets.** These datasets are generated from the Stochastic Block Model (SBM). We use the model to plant ground-truth clusters. It samples positive edges between vertex pairs within the same planted cluster with probability $p$, and samples positive edges across different clusters with probability $(1 - p)$. In the main text, we set $p = 0.95$. **2) Real-world datasets.** We use EMAILCORE (Leskovec et al., 2007; Yin et al., 2017), FACEBOOK (McAuley & Leskovec, 2012) and LASTFM (Rozemberczki & Sarkar, 2020) datasets. We refer to Appendix G.1 for detailed descriptions. For simplicity, for all datasets, we only simulate insertion-only streams of edges.

**Predictor description. 1) Noisy predictor.** We use this predictor for datasets with available optimal clusterings. We form this predictor by performing perturbations on optimal clusterings. **2) Spectral clustering.** We use this predictor for EMAILCORE and LASTFM. It first maps all the vertices to a $d$-dimensional Euclidean space using the graph Laplacian, then clusters all the vertices based on their embeddings. For any two vertices $u, v \in V$, we form the prediction $d_{uv}$ based on the spectral embeddings of $u$ and $v$. We refer to Appendix G.2 for detailed descriptions.

**Baselines. 1)** $(3 + \varepsilon)$**-approximation non-learning counterparts.** For our algorithm in dynamic streams, the counterpart is Algorithm CKLPU24 (Cambus et al., 2024); for insertion-only streams, the counterpart is Algorithm CM23 (Chakrabarty & Makarychev, 2023). **2) The agreement decomposition algorithm** CLMNPT21 (Cohen-Addad et al., 2021). Though the approximation ratio

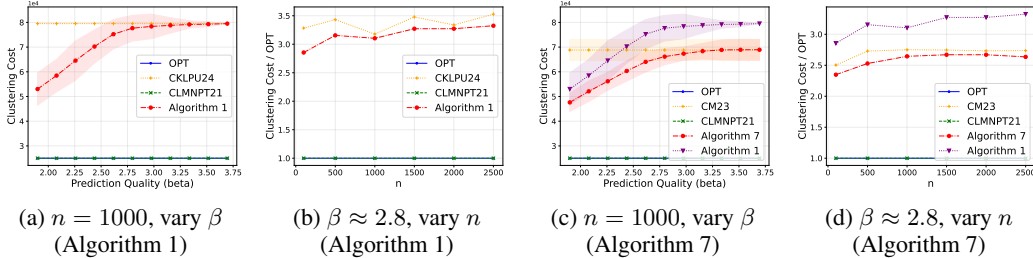

(a) $n = 1000$, vary $\beta$
(Algorithm 1)

(b) $\beta \approx 2.8$, vary $n$
(Algorithm 1)

(c) $n = 1000$, vary $\beta$
(Algorithm 7)

(d) $\beta \approx 2.8$, vary $n$
(Algorithm 7)

Figure 1: Performance of our algorithms on synthetic datasets with SBM parameter $p = 0.95$. We examine of the effect of prediction quality $\beta$ and graph size $n$. We set $k = 10$ for Algorithm 7.

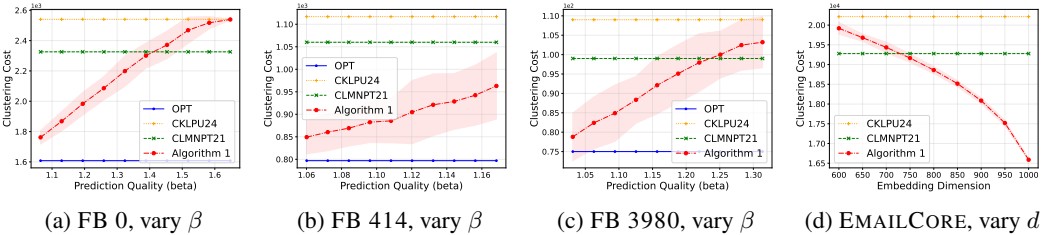

(a) FB 0, vary $\beta$

(b) FB 414, vary $\beta$

(c) FB 3980, vary $\beta$

(d) EMAILCORE, vary $d$

Figure 2: Performance of Algorithm 1 on real-world datasets. Figures 2(a)–(c) show the effect of $\beta$ on FACEBOOK subgraphs. Figure 2(d) shows the effect of the dimension $d$ of spectral embeddings on EMAILCORE. Note that a larger $d$ indicates higher prediction quality (i.e., a smaller $\beta$).

in theory is large ($\approx 701$), this algorithm has been shown to give high-quality solutions in practice. Note that this algorithm only works for insertion-only streams and requires multiple passes. For a fair comparison, we ensured that all baselines were implemented with equal effort.

**Results on synthetic datasets.** Figure 1 shows the performance of our algorithms on synthetic datasets. **1) Varying $\beta$.** We first examine the effect of $\beta$ (see Figures 1(a) and (c)). We can see that when $\beta$ is small, the cost of our algorithms is significantly lower than that of the $(3 + \varepsilon)$-approximation non-learning counterparts. Even when $\beta$ is large, our algorithms do not perform worse than theirs. Notably, we observe that the algorithm of CLMNPT21 outputs the optimal solution. We attribute this to the fact that the SBM graphs contain many dense components, which makes them well-suited for the algorithm. **2) Varying $n$.** Furthermore, we investigate whether our algorithms scale well with graph size (see Figures 1(b) and (d)). To clearly present our results, we calculate the ratio between the cost of each algorithm and the optimal solution. The result demonstrates that our algorithms perform well consistently as the graph size increases.

**Results on real-world datasets.** Figure 2 shows the performance of our algorithm in dynamic streams (Algorithm 1) on real-world datasets. The results demonstrate that under good prediction quality, Algorithm 1 consistently outperforms other baselines across all datasets used. For example, in Figure 2(a), when $\beta \approx 1.2$, the average cost of our algorithm is $15\%$ lower than that of CLM-NPT21 and $22\%$ lower than that of CKLPU24. Besides, in Figure 2(d), our algorithm reduces the clustering cost by up to $14\%$ compared to CLMNPT21. Even in case of poor predictions, Algorithm 1 does not perform worse than the $(3 + \varepsilon)$-approximation counterparts without predictions.

## 6 CONCLUSION

We present the first LAAs for Correlation Clustering in the streaming setting by leveraging $\beta$-level predictions. Specifically, we provide single-pass streaming algorithms that achieve a $(\min\{2.06\beta, 3\} + \varepsilon)$-approximation for Correlation Clustering in both insertion-only and dynamic streams. In particular, our algorithm in the dynamic setting is the first better-than-3-approximation algorithm for Correlation Clustering in this context. Additionally, our algorithm is quite simple and easy for implementation. There are many interesting future research directions, such as achieving better space-approximation trade-offs with predictions than the standard setting, and finding more applications of prediction-based graph sparsification or sampling.

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

# A OTHER RELATED WORK

**Correlation Clustering.** In this paper, we focus on the minimization version of Correlation Clustering (i.e., minimizing the number of disagreements), which is the most commonly studied version. There are other variants of this problem. For example, Swamy (2004) discussed the maximization version, which is to maximize the number of agreements, and provided a $0.766$-approximation algorithm via SDP. This problem is further examined on general graphs (Charikar et al., 2005) and on weighted graphs (Demaine et al., 2006).

**Learning-Augmented Algorithms.** Learning-augmented algorithms (LAAs; also known as algorithms with predictions) have been actively researched in online algorithms (Purohit et al., 2018; Bamas et al., 2020; Lattanzi et al., 2020; Im et al., 2021; Lykouris & Vassilvitskii, 2021; Antoniadis et al., 2023a;b; Angelopoulos et al., 2024), data structures (Mitzenmacher, 2018; Ferragina & Vinciguerra, 2020; Vaidya et al., 2021; Lin et al., 2022; Sato & Matsui, 2023), graph algorithms (Dinitz et al., 2021; Chen et al., 2022b; Banerjee et al., 2023; Lattanzi et al., 2023; Davies et al., 2023; Liu & Srinivas, 2023; Brand et al., 2024; Henzinger et al., 2024; DePavia et al., 2024), sublinear-time algorithms (Indyk et al., 2019; Eden et al., 2021; Li et al., 2023), and approximation algorithms (Ergun et al., 2022; Nguyen et al., 2023; Cohen-Addad et al., 2024a; Ghoshal et al., 2024; Braverman et al., 2024). In this paper, we focus on learning-augmented algorithms in the graph streaming model.

# B USEFUL TOOLS

Our algorithms use the graph sparsification techniques, so we need the following definitions.

**Definition B.1** ($\ell_0$-sampler (Jowhari et al., 2011)). Let $\boldsymbol{x} \in \mathbb{R}^n$ be a non-zero vector and $\delta \in (0, 1)$. An $\ell_0$-sampler for $\boldsymbol{x}$ returns FAIL with probability at most $\delta$ and otherwise returns some index $i$ such that $x_i \neq 0$ with probability $\frac{1}{|\operatorname{supp}(\boldsymbol{x})|}$ where $\operatorname{supp}(\boldsymbol{x}) = \{i \mid x_i \neq 0\}$ is the support of $\boldsymbol{x}$.

The following theorem states that $\ell_0$-samplers can be maintained using a single pass in dynamic streams.

**Theorem B.2** (Jowhari et al. (2011)). *There exists a single-pass streaming algorithm for maintaining an $\ell_0$-sampler for a non-zero vector $\boldsymbol{x} \in \mathbb{R}^n$ (with failure pribability $\delta$) in the dynamic model using $O(\log^2 n \log \delta^{-1})$ bits of space.*

We can use $\ell_0$-samplers to construct graph sparsifiers.

**Definition B.3** (Cut sparsifier). Let $H = (V_H, E_H)$ be an undirected unweighted (but not necessarily complete) graph and $\varepsilon \in (0, 1)$, we say that a weighted subgraph $H' = (V_H, E'_H, w)$ is an $\varepsilon$-cut-sparsifier of $H$ if for any $A \subseteq V_H$,

$$(1 - \varepsilon)\delta_H(A) \leq \delta_{H'}(A) \leq (1 + \varepsilon)\delta_H(A),$$

where $\delta_H(A) := |\{(u, v) \mid u \in A, v \in V_H \setminus A\}|$ denotes the *size* of the cut $(A, V_H \setminus A)$ in $H$, and $\delta_{H'}(A) := \sum_{e \in C} w_e$, where $C = \{(u, v) \mid u \in A, v \in V_H \setminus A\}$, denotes the *weight* of the cut $(A, V_H \setminus A)$ in $H'$.

**Definition B.4** (Spectral sparsifier). Let $H = (V_H, E_H)$ be an undirected unweighted (but not necessarily complete) graph and $\varepsilon \in (0, 1)$, we say that a weighted subgraph $H' = (V_H, E'_H, w)$ is an $\varepsilon$-spectral-sparsifier of $H$ if for any $\boldsymbol{x} \in \mathbb{R}^n$,

$$(1 - \varepsilon)\boldsymbol{x}^\top L_H \boldsymbol{x} \leq \boldsymbol{x}^\top L_{H'} \boldsymbol{x} \leq (1 + \varepsilon)\boldsymbol{x}^\top L_H \boldsymbol{x},$$

which is equivalent to

$$(1 - \varepsilon)L_H \preceq L_{H'} \preceq (1 + \varepsilon)L_H,$$

where $L_H$ is the Laplacian of $H$, and $L_{H'}$ is the Laplacian of $H'$.

It is easy to see that if $H'$ is an $\varepsilon$-spectral-sparsifier of $H$, then $H'$ is also an $\varepsilon$-cut-sparsifier of $H$.

The following theorem states that a spectral sparsifier can be constructed using a single pass and $O(\varepsilon^{-2} n \log n)$ space in insertion-only streams.

**Theorem B.5** (Kelner & Levin (2011)). *There exists a single-pass streaming algorithm for constructing an $\varepsilon$-spectral-sparsifier of an unweighted, undirected graph in the insertion-only model using $O(\varepsilon^{-2} n \log n)$ space. The algorithm succeeds with high probability.*

Since a spectral sparsifier implies a cut sparsifier, we can construct a cut sparsifier using a single pass and $O(\varepsilon^{-2} n \log n)$ space in insertion-only streams. Let KL-SPARSIFICATION be any algorithm for constructing a cut sparsifier that satisfies the above guarantees.

The following theorem states that a cut sparsifier can be constructed using a single pass and $\widetilde{O}(\varepsilon^{-2} n)$ space in dynamic streams.

**Theorem B.6** (Ahn et al. (2012)). *There exists a single-pass streaming algorithm for constructing an $\varepsilon$-cut-sparsifier of an unweighted, undirected graph in the dynamic model using $O(n \log^6 n + \varepsilon^{-2} n \log^5 n)$ space. The algorithm succeeds with high probability.*

Let AGM-SPARSIFICATION be any algorithm for constructing a cut sparsifier that satisfies the guarantees of Theorem B.6.

## C    OMITTED PSEUDOCODES OF SECTION 3

In this section, we give the omitted pseudocodes of Section 3: Algorithm 2, Algorithm 3, Algorithm 4, Algorithm 5 and Algorithm 6.

---

**Algorithm 2** Offline implementation of our algorithm in dynamic streams

---

**Input:** Graph $G^+ = (V, E^+)$, oracle access to a $\beta$-level predictor $\Pi$
**Output:** Partition of $V$ into disjoint sets
 1: Pick a random permutation of vertices $\pi : V \to \{1, \ldots, n\}$.
 2: Initially, all vertices are unclustered and interesting.
 3: A vertex $u$ marks itself uninteresting if $\pi_u \geq \tau_u$ where $\tau_u := \frac{c}{\varepsilon} \cdot \frac{n \log n}{\deg^+(u)}$.
 4: Let $G_{\text{store}}$ be the graph induced by the interesting vertices.
 5: $\mathcal{C}_1 \leftarrow$ TRUNCATEDPIVOT$(G^+, G_{\text{store}}, \pi)$
 6: $\mathcal{C}_2 \leftarrow$ TRUNCATEDPIVOTWITHPRED$(G^+, G_{\text{store}}, \pi, \Pi)$
 7: $i \leftarrow \arg \min_{i=1,2} \{\text{cost}_G(\mathcal{C}_i)\}$
 8: **return** $\mathcal{C}_i$

---

**Algorithm 3** TRUNCATEDPIVOT$(G^+, H, \pi)$

---

**Input:** Graph $G^+ = (V, E^+)$, induced subgraph $H = (V_H, E_H)$ where $V_H \subseteq V$ and $E_H \subseteq E^+$, permutation $\pi : V \to \{1, \ldots, n\}$
**Output:** Partition of $V$ into disjoint sets
 1: Let $U^{(1)} \leftarrow V_H$ be the set of unclustered vertices in $V_H$.
 2: Let $t \leftarrow 1$.
 3: **while** $U^{(t)} \neq \emptyset$ **do**
 4:     Let $u \in U^{(t)}$ be the vertex with the smallest rank.
 5:     Mark $u$ as a pivot. Initialize a new *pivot cluster* $S^{(t)} \leftarrow \{u\}$.
 6:     For each vertex $v \in U^{(t)}$ such that $(u, v) \in E_H$, add $v$ to $S^{(t)}$.
 7:     Remove all vertices clustered at this iteration from $U^{(t)}$.
 8:     $t \leftarrow t + 1$.
 9: Each vertex $u \in V \setminus V_H$ joins the cluster of pivot $v$ with the smallest rank, if $(u, v) \in E^+$ and $\pi_v < \tau_u$.
10: Each unclustered vertex $u \in V$ creates a *singleton cluster*.
11: **return** the final clustering $\mathcal{C}$, which contains all pivot clusters and singleton clusters

---

---

**Algorithm 4** TRUNCATEDPIVOTWITHPRED$(G^+, H, \pi, \Pi)$

---

**Input:** Graph $G^+ = (V, E^+)$, induced subgraph $H = (V_H, E_H)$ where $V_H \subseteq V$ and $E_H \subseteq E^+$, permutation $\pi : V \to \{1, \ldots, n\}$, oracle access to a $\beta$-level predictor $\Pi$
**Output:** Partition of $V$ into disjoint sets
 1: Let $U^{(1)} \leftarrow V_H$ be the set of unclustered vertices in $V_H$.
 2: Let $t \leftarrow 1$.
 3: For any $u, v \in V$, $d_{uv} = \Pi(u, v)$.
 4: For any $u, v \in V$, define $p_{uv} := f(d_{uv})$.
 5: **while** $U^{(t)} \neq \emptyset$ **do**
 6:     Let $u \in U^{(t)}$ be the vertex with the smallest rank.
 7:     Mark $u$ as a pivot. Initialize a new *pivot cluster* $S^{(t)} \leftarrow \{u\}$.
 8:     For each vertex $v \in U^{(t)}$, add $v$ to $S^{(t)}$ with probability $(1 - p_{uv})$ independently.
 9:     Remove all vertices clustered at this iteration from $U^{(t)}$.
10:     $t \leftarrow t + 1$.
11:  Each vertex $u \in V \setminus V_H$ joins the cluster of pivot $v$ in the order of $\pi$ with probability $(1 - p_{uv})$ independently, if $\pi_v < \tau_u$.
12: Each unclustered vertex $u \in V$ creates a *singleton cluster*.
13: **return** the final clustering $\mathcal{C}$, which contains all pivot clusters and singleton clusters

---

**Algorithm 5** CKLPU-PIVOT$(G^+)$

---

**Input:** Graph $G^+ = (V, E^+)$
**Output:** Partition of vertices into disjoint sets
 1: Pick a random permutation of vertices $\pi : V \to \{1, \ldots, n\}$.
 2: Let $U^{(1)} \leftarrow V$ be the set of unclustered vertices.
 3: **for** $t = 1, \ldots, n$ **do**
 4:     Let $\ell_t \leftarrow \frac{c}{\varepsilon} \cdot \frac{n \log n}{t}$.
 5:     Let $u \in V$ be the $t$-th vertex in $\pi$ (i.e., $t = \pi_u$).
 6:     Each unclustered vertex $v$ with $\deg^+(v) \geq \ell_t$ creates a *singleton cluster*.
 7:     **if** $u \in U^{(t)}$ **then**
 8:         Mark $u$ as a pivot. Initialize a new *pivot cluster* $S^{(t)} \leftarrow \{u\}$.
 9:         For each vertex $v \in N^+(u) \cap U^{(t)}$, add $v$ to $S^{(t)}$.
10:     Remove all vertices clustered at this iteration from $U^{(t)}$.
11: **return** the final clustering $\mathcal{C}$, which contains all pivot clusters and singleton clusters

---

**Algorithm 6** PAIRWISEDISS$(G^+, \Pi)$

---

**Input:** Graph $G^+ = (V, E^+)$, oracle access to a $\beta$-level predictor $\Pi$
**Output:** Partition of vertices into disjoint sets
 1: Pick a random permutation of vertices $\pi : V \to \{1, \ldots, n\}$.
 2: For any $u, v \in V$, $d_{uv} = \Pi(u, v)$.
 3: For any $u, v \in V$, define $p_{uv} := f(d_{uv})$.
 4: Let $U^{(1)} \leftarrow V$ be the set of unclustered vertices.
 5: **for** $t = 1, \ldots, n$ **do**
 6:     Let $\ell_t \leftarrow \frac{c}{\varepsilon} \cdot \frac{n \log n}{t}$.
 7:     Let $u \in V$ be the $t$-th vertex in $\pi$ (i.e., $t = \pi_u$).
 8:     Each unclustered vertex $v$ with $\deg^+(v) \geq \ell_t$ creates a *singleton cluster*.
 9:     **if** $u \in U^{(t)}$ **then**
10:         Mark $u$ as a pivot. Initialize a new *pivot cluster* $S^{(t)} \leftarrow \{u\}$.
11:         For each vertex $v \in U^{(t)}$, add $v$ to $S^{(t)}$ with probability $(1 - p_{uv})$ independently.
12:     Remove all vertices clustered at this iteration from $U^{(t)}$.
13: **return** the final clustering $\mathcal{C}$, which contains all pivot clusters and singleton clusters

---

# D OMITTED PROOFS OF SECTION 3

## D.1 PROOF OF THEOREM 1.1

**Space Complexity.** We first analyze the space complexity of Algorithm 1. For each vertex $u \in V$, we mainly store its rank $\pi_u$, positive degree $\deg^+(u)$, and $10c \log n \cdot \sigma_u$ independent $\ell_0$-samplers. We have the following lemma which states the space requirement of $\ell_0$-samplers.

**Lemma D.1** (Cambus et al. (2024)). *The $\ell_0$-samplers used in Algorithm 1 require $O(\varepsilon^{-1} n \log^4 n)$ words of space.*

Furthermore, by Theorem B.6, the AGM-SPARSIFICATION algorithm uses $O(n \log^6 n + \varepsilon^{-2} n \log^5 n)$ words of space. Therefore, the space complexity of Algorithm 1 is $O(n \log^6 n + \varepsilon^{-2} n \log^5 n)$ words.

**Approximation Guarantee.** Next, we analyze the approximation ratio of Algorithm 1. We rely on the following lemma.

**Lemma D.2** (Lemma 2 in Cambus et al. (2024)). *The $\ell_0$-samplers allow us to recover the positive edges incident to all interesting vertices with high probability.*

Therefore, Algorithm 1 works with the same set of edges as Algorithm 2 in the clustering phase with high probability. This implies that both algorithms return the same clustering with the same probability. On the other hand, if the high probability event of Lemma D.2 does not happen, then Algorithm 1 produces a clustering of cost at most $O(n^2)$, which leads to an additive $1/\operatorname{poly}(n)$ term to the expected cost of Algorithm 1 compared to that of Algorithm 2. This preserves the approximation ratio if $\mathrm{OPT} \neq 0$.

We also need the following lemma which shows that the estimate $\widetilde{\operatorname{cost}}_G(\mathcal{C})$ well approximates the cost of any clustering $\mathcal{C}$ of $G$.

**Lemma D.3** (Behnezhad et al. (2023)). *Let $\varepsilon \in (0, 1)$. For any clustering $\mathcal{C}$ of $V$, the cost $\operatorname{cost}_G(\mathcal{C})$ is approximated by the estimate $\widetilde{\operatorname{cost}}_G(\mathcal{C}) := \sum_{C \in \mathcal{C}} \left( \frac{1}{2} \delta_{H^+}(C) + \binom{|C|}{2} - \frac{1}{2} \sum_{u \in C} \deg^+(u) \right)$ up to a multiplicative factor of $(1 \pm \varepsilon)$.*

Therefore, Theorem 1.1 follows from Lemma D.2 and Lemma D.3 by applying the union bound.

## D.2 PROOF OF LEMMA 3.3

The proof is similar to that of Lemma 3.2. The proof idea is as follows: we first show that in both cases, the singleton clusters $V_{\sin}$ are the same (with the same probability). Then we show that the randomized pivot-based algorithm runs on the same subgraph $G^+[V \setminus V_{\sin}]$ (with the same probability) in both cases, therefore outputting the same pivot clusters (with the same probability).

Consider a vertex $u$ that is unclustered at the beginning of iteration $t \ (\leq \pi_u)$, and becomes a singleton cluster due to Line 8 of Algorithm PAIRWISEDISS. By definition, $t$ is the smallest integer such that $\deg^+(u) \geq \frac{c}{\varepsilon} \cdot \frac{n \log n}{t}$ and hence $t = \lceil \tau_u \rceil$. Since $t \leq \pi_u$, we have $\deg^+(u) \geq \frac{c}{\varepsilon} \cdot \frac{n \log n}{\pi_u}$, which corresponds to $u$ becoming uninteresting in Algorithm 2. Since $u$ is in a singleton cluster, it did not join any pivot cluster, implying that for any vertex $v \neq u$, either (1) $\pi_v \geq t$, or (2) the event that $v$ is a pivot and $u$ joins the cluster of $v$ satisfying $\pi_v < t$ does not happen. This is equivalent to saying that the event that $u$ joins the cluster of pivot $v$ satisfying $\pi_v < \tau_u$ does not happen, since $\pi_v$ is an integer. By Line 11 of Algorithm TRUNCATEDPIVOTWITHPRED, $u$ creates a singleton cluster in Line 6 of Algorithm 2 (with the same probability) as well.

Now consider a vertex $u$ that creates a singleton cluster in Line 6 of Algorithm 2. Then $u$ must be marked uninteresting (implying $\pi_u \geq \tau_u$), and $u$ can neither be a pivot nor join the cluster of pivot $v$ satisfying $\pi_v < \tau_u$. By definition of $\tau_u$, iteration $\lceil \tau_u \rceil$ is the smallest iteration such that $\deg^+(u) \geq \frac{c}{\varepsilon} \cdot \frac{n \log n}{\lceil \tau_u \rceil}$. This implies that $u$ is unclustered at the beginning of iteration $\lceil \tau_u \rceil$ in Algorithm PAIRWISEDISS, and forms a singleton cluster in that iteration (with the same probability).

Since the vertices forming singleton clusters are the same in both cases (with the same probability), the subgraph induced by the remaining vertices $G^+[V \setminus V_{\sin}]$ is the same (with the same probability).

The same randomized pivot-based algorithm runs on $G^+[V \setminus V_{\text{sin}}]$ in both cases, which implies that the pivots will be the same (with the same probability). Finally, we observe that in both cases, a non-pivot vertex $u$ joins the cluster of pivot $v$ such that $\pi_v < \tau_u$ in the order of $\pi$ with probability $(1 - p_{uv})$ independently. Hence, the pivot clusters are the same (with the same probability).

### D.3 PROOF OF COROLLARY 3.7

Corollary 3.7 follows from Lemma 3.3, Lemma 3.4 and Lemma 3.6.

## E OMITTED PSEUDOCODES OF SECTION 4

In this section, we give the omitted pseudocodes of Section 4: Algorithm 7, Algorithm 8, Algorithm 9, Algorithm 10 and Algorithm 11.

---

**Algorithm 7** An insertion-only streaming algorithm for Correlation Clustering with predictions

---

**Input:** Complete graph $G = (V, E = E^+ \cup E^-)$ as an arbitrary-order stream of edges, oracle access to a $\beta$-level predictor $\Pi$, integer $k$
**Output:** Partition of $V$ into disjoint sets
    ▷ **Preprocessing phase**
1: Pick a random permutation of vertices $\pi : V \to \{1, \dots, n\}$.
2: For any $u, v \in V$, $d_{uv} = \Pi(u, v)$.
3: For any $u, v \in V$, define $p_{uv} := f(d_{uv})$.
4: **for** each vertex $u \in V$ **do**
5:     Create a priority queue $A(u)$ with a maximum size of $k$ and initialize $A(u) \leftarrow \{u\}$.
6:     Create a priority queue $B(u)$ with a maximum size of $k$ and initialize $B(u) \leftarrow \{u\}$.
7:     $\deg^+(u) \leftarrow 0$
8: Initialize a cut sparsifier $H^+$ for the subgraph $G^+ = (V, E^+)$.
    ▷ **Streaming phase**
9: **for** each edge $e = (u, v) \in E$ **do**
10:     **if** $e = (u, v) \in E^+$ **then**
11:         Add $u$ to $A(v)$. Add $v$ to $A(u)$.
12:         **if** $|A(u)| > k$ (resp. $|A(v)| > k$) **then**
13:             Remove the vertex with the highest rank from $A(u)$ (resp. $A(v)$).
14:         $\deg^+(u) \leftarrow \deg^+(u) + 1, \deg^+(v) \leftarrow \deg^+(v) + 1$
15:         Apply KL-SPARSIFICATION$(H^+, e)$.
16:     With probability $(1 - p_{uv})$, add $u$ to $B(v)$ and add $v$ to $B(u)$.
17:     **if** $|B(u)| > k$ (resp. $|B(v)| > k$) **then**
18:         Remove the vertex with the highest rank from $B(u)$ (resp. $B(v)$).
    ▷ **Postprocessing phase**
19: $\mathcal{C}_1 \leftarrow$ CLUSTER$(V, \pi, \{A(u)\}_{u \in V})$
20: $\mathcal{C}_2 \leftarrow$ CLUSTER$(V, \pi, \{B(u)\}_{u \in V})$
21: $\widetilde{W}_1 \leftarrow$ ESTIMATECOST$(\mathcal{C}_1, \{\deg^+(u)\}_{u \in V}, H^+)$
22: $\widetilde{W}_2 \leftarrow$ ESTIMATECOST$(\mathcal{C}_2, \{\deg^+(u)\}_{u \in V}, H^+)$
23: $i \leftarrow \arg\min_{i=1,2}\{\widetilde{W}_i\}$
24: **return** $\mathcal{C}_i$

---

## F OMITTED DETAILS OF SECTION 4

### F.1 OUR ALGORITHM IN INSERTION-ONLY STREAMS

Recall that we have oracle access to a $\beta$-level predictor $\Pi$, which can predict the pairwise dissimilarity $d_{uv} \in [0, 1]$ between any two vertices $u$ and $v$ in $G$.

Based on the predictions, we propose a single-pass semi-streaming algorithm which works in insertion-only streams (see Algorithm 7). We first pick a random permutation of vertices $\pi : V \to$

---

**Algorithm 8** CLUSTER$(V, \pi, \{T(u)\}_{u \in V})$

---

**Input:** Vertex set $V$, permutation of vertices $\pi : V \to \{1, \ldots, n\}$, truncated neighbors of each vertex $\{T(u)\}_{u \in V}$
**Output:** Partition of $V$ into disjoint sets
1: **for** each unclustered vertex $u \in V$ chosen in the order of $\pi$ **do**
2:     Find the vertex $v \in T(u)$ with the lowest rank such that $v$ is a pivot or $v = u$, i.e., $v \leftarrow \arg\min_{v \in T(u)}\{\pi_v : v \text{ is a pivot or } v = u\}$.
3:     **if** such a vertex $v$ exists **then**
4:         Put $u$ in the cluster of $v$.
5:         **if** $v = u$ **then**
6:             Mark $u$ as a *pivot*.
7:         **else**
8:             Put $u$ in a singleton cluster. Mark $u$ as a *singleton*.
9: **return** the final clustering $\mathcal{C}$

---

**Algorithm 9** CM-PIVOT$(G, k)$

---

**Input:** Complete graph $G = (V, E = E^+ \cup E^-)$, integer $k$
**Output:** Partition of vertices into disjoint sets
1: Let $F^{(1)} \leftarrow V$ be the set of fresh vertices.
2: Let $U^{(1)} \leftarrow V$ be the set of unclustered vertices.
3: For each vertex $u \in V$, initialize a counter $K^{(1)}(u) \leftarrow 0$.
4: Let $t \leftarrow 1$.
5: **while** $F^{(t)} \neq \emptyset$ **do**
6:     Choose a vertex $w^{(t)} \in F^{(t)}$ uniformly at random.
7:     **if** $w^{(t)} \in U^{(t)}$ **then**
8:         Mark $w^{(t)}$ as a pivot. Initialize a new pivot cluster $S^{(t)} \leftarrow \{w^{(t)}\}$.
9:         For each vertex $v \in N^+(w^{(t)}) \cap U^{(t)}$, add $v$ to $S^{(t)}$.
10:    **else**
11:        For each vertex $v \in N^+(w^{(t)}) \cap U^{(t)}$, let $K^{(t+1)}(v) \leftarrow K^{(t)}(v) + 1$. Subsequently, all vertices $v$ with $K^{(t+1)}(v) = k$ are put into singleton clusters.
12:    Let $F^{(t+1)} \leftarrow F^{(t)} \setminus \{w^{(t)}\}$ and remove all vertices clustered at this iteration from $U^{(t)}$.
13:    Let $t \leftarrow t + 1$.
14: **return** the final clustering $\mathcal{C}$, which contains all pivot clusters and singleton clusters

---

$\{1, \ldots, n\}$. For each vertex $u \in V$, we initialize two priority queues $A(u)$ and $B(u)$, each with a maximum size capped at $k$, where $k$ is a constant. Initially, we add $u$ to both queues. During the streaming phase, we employ two distinct methods to retain at most $k$ neighbors of each vertex. Specifically, for each edge $(u, v) \in E$ in the stream, if $(u, v)$ is a positive edge, we add $u$ to $A(v)$ and add $v$ to $A(u)$. Additionally, regardless of whether $(u, v)$ is positive or negative, we add $u$ to $B(v)$ with probability $(1 - p_{uv})$ and add $v$ to $B(u)$ with probability $(1 - p_{uv})$, where $p_{uv} = f(d_{uv})$ and $d_{uv} = \Pi(u, v)$. Note that if the size of any queue exceeds $k$, then we remove the vertex with the highest rank from the queue. That is, $A(u)$ maintains at most $k$ positive neighbors of $u$ with lowest ranks, while $B(u)$ contains at most $k$ neighbors (not necessarily positive) of $u$ with lowest ranks, the inclusion of which is probabilistic. Note that we define the rank of a vertex as its order in the permutation $\pi$, e.g., $\pi_u$ is the rank of $u$.

After the streaming phase, we run Algorithm 8 on the truncated graphs induced by both sets of priority queues, i.e., $\{A(u)\}_{u \in V}$ and $\{B(u)\}_{u \in V}$. Specifically, for each vertex $u$ picked in the order of $\pi$, we determine the cluster to which $u$ belongs. We try to find the vertex $v$ with the lowest rank in the queue of $u$, such that $v$ is a pivot or $v = u$. If such a vertex $v$ does not exist, then we mark $u$ as a singleton and place it in a singleton cluster. Otherwise, we assign $u$ to the cluster of $v$. Additionally, if $v = u$, then we mark $u$ as a pivot. Finally, we obtain two clusterings, each corresponding to a set of priority queues. We output the clustering with the lower cost.

It is worth noting that in the final step, the cost of a clustering cannot be exactly calculated, as our streaming algorithm cannot store the entire graph. To overcome this challenge, we utilize the graph

---

**Algorithm 10** PAIRWISEDISS2($G, \Pi, k$)

---

**Input:** Complete graph $G = (V, E = E^+ \cup E^-)$, oracle access to a $\beta$-level predictor $\Pi$, integer $k$
**Output:** Partition of vertices into disjoint sets
 1: Let $F^{(1)} \leftarrow V$ be the set of fresh vertices.
 2: Let $U^{(1)} \leftarrow V$ be the set of unclustered vertices.
 3: For each vertex $u \in V$, initialize a counter $K^{(1)}(u) \leftarrow 0$.
 4: For any $u, v \in V$, $d_{uv} = \Pi(u, v)$.
 5: For any $u, v \in V$, define $p_{uv} := f(d_{uv})$.
 6: Let $t \leftarrow 1$.
 7: **while** $F^{(t)} \neq \emptyset$ **do**
 8:     Choose a vertex $w^{(t)} \in F^{(t)}$ uniformly at random.
 9:     **if** $w^{(t)} \in U^{(t)}$ **then**
10:         Mark $w^{(t)}$ as a pivot. Initialize a new pivot cluster $S^{(t)} \leftarrow \{w^{(t)}\}$.
11:         For each vertex $v \in U^{(t)}$, add $v$ to $S^{(t)}$ with probability $(1 - p_{vw^{(t)}})$ independently.
12:     **else**
13:         For each vertex $v \in U^{(t)}$, let $K^{(t+1)}(v) \leftarrow K^{(t)}(v) + 1$ with probability $(1 - p_{vw^{(t)}})$ independently. Subsequently, all vertices $v$ with $K^{(t+1)}(v) = k$ are put into singleton clusters.
14:     Let $F^{(t+1)} \leftarrow F^{(t)} \setminus \{w^{(t)}\}$ and remove all vertices clustered at this iteration from $U^{(t)}$.
15:     Let $t \leftarrow t + 1$.
16: **return** the final clustering $\mathcal{C}$, which contains all pivot clusters and singleton clusters

---

**Algorithm 11** PAIRWISEDISS2WITHPREROUNDING($G, \Pi, k$)

---

**Input:** Complete graph $G = (V, E = E^+ \cup E^-)$, oracle access to a $\beta$-level predictor $\Pi$, integer $k$
**Output:** Partition of vertices into disjoint sets
 1: For any $u, v \in V$, $d_{uv} = \Pi(u, v)$.
 2: For any $u, v \in V$, define $p_{uv} := f(d_{uv})$.
 3: $E'^{+} \leftarrow \emptyset$.
 4: **for** each edge $(u, v) \in E$ such that $p_{uv} < 1$ **do**
 5:     add $(u, v)$ to $E'^{+}$ with probability $(1 - p_{uv})$.
 6: $E'^{-} \leftarrow E \setminus E'^{+}$
 7: $\mathcal{C} \leftarrow$ CM-PIVOT($G' := (V, E'^{+} \cup E'^{-}), k$)
 8: **return** $\mathcal{C}$

---

sparsification technique (Kelner & Levin, 2011) to estimate the cost of a clustering. During the streaming phase, we maintain a cut sparsifier $H^+$ for the subgraph $G^+ = (V, E^+)$. Specifically, for each positive edge $(u, v) \in E^+$ in the stream, we apply KL-SPARSIFICATION($H^+, (u, v)$) to determine whether $(u, v)$ is added to $H^+$ and, if so, its corresponding weight in $H^+$. We also maintain the positive degree $\deg^+(u)$ of each vertex $u$. According to Theorem B.5, the sparsifier can be constructed using a single pass and can approximate the value of every cut in $G^+$ up to a $(1 \pm \varepsilon)$-multiplicative error with high probability. Thus we can to approximate the cost of a clustering using the stored information up to a $(1 \pm \varepsilon)$-multiplicative error with high probability, by the guarantee of (Behnezhad et al., 2023).

### F.2 ANALYSIS

#### F.2.1 SPACE COMPLEXITY

For each vertex $u \in V$, we mainly store its rank $\pi_u$, positive degree $\deg^+(u)$, and at most $2k$ vertices. As we will see, we set $k = O(1/\varepsilon)$. Furthermore, by Theorem B.5, the KL-SPARSIFICATION algorithm uses $O(\varepsilon^{-2} n \log n)$ words of space. Therefore, the total space complexity of the algorithm is $O(\varepsilon^{-2} n \log n)$ words.

#### F.2.2 ALGORITHM 7 AS A COMBINATION OF ALGORITHMS CM-PIVOT AND PAIRWISEDISS2

We define a permutation $\pi$ for Algorithms CM-PIVOT and PAIRWISEDISS2 as $\pi : w^{(t)} \mapsto t$, where $w^{(t)}$ is the vertex picked at iteration $t$ of Algorithms CM-PIVOT and PAIRWISEDISS2. Obviously,

$\pi$ is a uniformly random permutation over $V$. Therefore, we can also view Algorithms CM-PIVOT and PAIRWISEDISS2 from an equivalent perspective: at the beginning of each iteration $t$, choose a vertex $w^{(t)}$ in the order of $\pi$. We have the following lemmas.

**Lemma F.1** (Lemma 2.1 in Chakrabarty & Makarychev (2023)). *If Algorithm 7 and Algorithm* CM-PIVOT *use the same permutation* $\pi$, *then Algorithm* CM-PIVOT *and Line 19 of Algorithm 7 output the same clustering of* $V$.

**Lemma F.2.** *If Algorithm 7 and Algorithm* PAIRWISEDISS2 *use the same permutation* $\pi$ *and predictions* $\{d_{uv}\}_{u,v \in V}$, *then Algorithm* PAIRWISEDISS2 *and Line 20 of Algorithm 7 output the same clustering of* $V$ *with the same probability.*

*Proof.* The proof is similar to that of Lemma F.1. Suppose that Algorithm 7 and Algorithm PAIRWISEDISS2 use the same permutation $\pi$ and predictions $\{d_{uv}\}_{u,v \in V}$, we want to prove that for each vertex $u \in V$, with the same probability, in both clusterings returned by Algorithm PAIRWISEDISS2 and Line 20 of Algorithm 7, $u$ is either assigned to the same pivot, or $u$ is placed into a singleton cluster.

We prove by induction on the rank $\pi_u$. Suppose that all vertices $v$ with $\pi_v < \pi_u$ are clustered in the same way with the same probability. If $u$ is put into a singleton cluster in the clustering returned by Line 20 of Algorithm 7, then there must exist $k$ vertices added to $B(u)$ probabilistically, and their ranks are lower than $\pi_u$. None of the vertices in $B(u)$ are pivots. Since both algorithms use the same $\pi$ and $\{d_{uv}\}_{u,v \in V}$, in Algorithm PAIRWISEDISS2, these $k$ vertices will cause the counter of $u$ to increment $k$ times probabilistically. Therefore, $u$ is also placed in a singleton cluster in the clustering returned by Algorithm PAIRWISEDISS2. And vice versa.

In Algorithm 7, if there are any pivots in $B(u)$ (or $u$ itself), then $u$ will be assigned to the pivot with the lowest rank (denoted as $v$). We have $\pi_v \leq \pi_u$ and $v$ has been added to $B(u)$ probabilistically. In Algorithm PAIRWISEDISS2, with the same probability, $v$ is marked as a pivot and $u$ is added to the cluster of $v$. And vice versa.

Therefore, Algorithm PAIRWISEDISS2 and Line 20 of Algorithm 7 cluster $u$ in the same way with the same probability. $\qquad\square$

### F.2.3 THE APPROXIMATION RATIOS OF CM-PIVOT AND PAIRWISEDISS2

In order to analyze the approximation ratio of Algorithm 7, it suffices to analyze Algorithms CM-PIVOT and PAIRWISEDISS2 respectively. We follow the analysis framework in Chakrabarty & Makarychev (2023). We categorize all iterations into *pivot iterations* and *singleton iterations*. Both iterations create some clusters. Consider iteration $t$ of both algorithms. If $w^{(t)} \in U^{(t)}$, then iteration $t$ is a pivot iteration; otherwise, it is a singleton iteration. We say that an edge $(u, v)$ is *decided* at iteration $t$ if both $u$ and $v$ were not clustered at the beginning of iteration $t$ (i.e., $u, v \in U^{(t)}$) but at least one of them was clustered at iteration $t$. Once an edge $(u, v)$ is decided, we can determine whether it contributes to the cost of the algorithm (i.e., the number of disagreements). Specifically, if $(u, v) \in E^+$, then it contributes to the cost of the algorithm if exactly one of $u$ and $v$ is assigned to the newly created cluster $S^{(t)}$; if $(u, v) \in E^-$, then it contributes to the cost of the algorithm if both $u$ and $v$ are assigned to the newly created cluster $S^{(t)}$.

Let $E^{(t)}$ denote the set of decided edges at pivot iteration $t$. Specifically, $E^{(t)} = \{(u, v) \mid u, v \in U^{(t)}; u \in S^{(t)} \text{ or } v \in S^{(t)}\}$. Let $P^{(t)}$ denote the cost of decided edges at pivot iteration $t$. We call the clusters created in pivot iterations *pivot clusters*. Let $P$ denote the cost of all pivot clusters. Therefore, $P = \sum_{t \text{ is a pivot iteration}} P^{(t)}$. Let $S$ denote the cost of all singleton clusters. Therefore, the cost of the algorithm is equal to $P + S$.

**Analysis of Algorithm CM-PIVOT.**

**Lemma F.3** (Chakrabarty & Makarychev (2023)). *Let $P_1$ denote the cost of pivot clusters returned by Algorithm* CM-PIVOT, *then* $\mathbb{E}[P_1] \leq 3 \cdot \text{OPT}$, *where* OPT *is the cost of the optimal solution on* $G$.

**Lemma F.4** (Chakrabarty & Makarychev (2023)). *Let $S_1$ denote the cost of singleton clusters returned by Algorithm* CM-PIVOT, *then* $\mathbb{E}[S_1] \leq \frac{6}{k-1} \cdot \text{OPT}$.

*Proof of Lemma 4.1.* Lemma 4.1 follows from Lemma F.1, Lemma F.3 and Lemma F.4. □

**Analysis of Algorithm PAIRWISEDISS2.**

*Proof of Lemma 4.2.* The key observation is that the pivot iterations in Algorithm PAIRWISEDISS2 are equivalent to the iterations of 2.06-approximation LP rounding algorithm by Chawla et al. (2015): given that $w^{(t)}$ is unclustered (i.e., $w^{(t)} \in U^{(t)}$), the conditional distribution of $w^{(t)}$ is uniformly distributed in $U^{(t)}$, and the cluster created during this iteration contains $w^{(t)}$ and all unclustered vertices $v$ added with probability $(1 - p_{vw^{(t)}})$. Therefore, we can directly apply the triangle-based analysis in (Chawla et al., 2015). Define $L := \sum_{(u,v) \in E^+} d_{uv} + \sum_{(u,v) \in E^-} (1 - d_{uv})$. Since the predictor is $\beta$-level, by Definition 2.1, we have that the predictions $\{d_{uv}\}_{u,v \in V}$ satisfy triangle inequality and $L \leq \beta \cdot \mathrm{OPT}$. It follows that for all pivot iterations $t$, $\mathbb{E}[P_2^{(t)}] \leq 2.06 \cdot \mathbb{E}[L^{(t)}]$, where $L^{(t)} := \sum_{(u,v) \in E^+ \cap E^{(t)}} d_{uv} + \sum_{(u,v) \in E^- \cap E^{(t)}} (1 - d_{uv})$. By linearity of expectation, we have $\mathbb{E}[P_2] = \mathbb{E}[\sum_{t \text{ is a pivot iteration}} P_2^{(t)}] = \sum_{t \text{ is a pivot iteration}} \mathbb{E}[P_2^{(t)}] \leq 2.06 \cdot L \leq 2.06\beta \cdot \mathrm{OPT}$. □

**Equivalence of Algorithms PAIRWISEDISS2 and PAIRWISEDISS2WITHPREROUNDING.**

**Claim F.5.** *If Algorithm* PAIRWISEDISS2 *and Algorithm* PAIRWISEDISS2WITHPREROUNDING *use the same permutation $\pi$ and predictions $\{d_{uv}\}_{u,v \in V}$, then they produce the same clustering with the same probability.*

*Proof.* The randomness in both algorithms comes from two sources: (1) the uniformly random permutation $\pi$ on vertices and (2) the probability that each vertex $v$ adjacent to $w^{(t)}$ will join the cluster of $w^{(t)}$ or increment its counter. The main difference between the two algorithms lies in the order in which the two sources of randomness are revealed: Algorithm PAIRWISEDISS2 can be viewed as choosing $\pi$ at the beginning and then performing iterations, where the randomness of all edges incident to $w^{(t)}$ is revealed after $w^{(t)}$ is chosen. In contrast, Algorithm PAIRWISEDISS2WITHPREROUNDING reveals the randomness of edges at the beginning, uses this information to construct a new instance, and then performs Algorithm CM-PIVOT on the new instance, where the randomness for $\pi$ is revealed. Note that the order of randomness does not affect the output. Therefore, if both algorithms use the same $\pi$ and $\{d_{uv}\}_{u,v \in V}$, then they will output the same clustering with the same probability. □

*Proof of Lemma 4.4.* By Lemma F.4, Claim F.5 and Lemma 4.3, we have $\mathbb{E}[S_2] \leq \frac{6}{k-1} \cdot \mathbb{E}[\mathrm{OPT}'] \leq \frac{6(2\beta+1)}{k-1} \cdot \mathrm{OPT}$. □

*Proof of Corollary 4.5.* Corollary 4.5 follows from Lemma F.2, Lemma 4.2 and Lemma 4.4. □

**Remark.** The reason our sampling-based approach works is mainly due to the fact that the rounding algorithm by Chawla et al. (2015) is equivalent to the algorithm that first samples a subgraph $G'$ according to the prediction oracle and then runs the PIVOT algorithm on $G'$. Therefore, if a Correlation Clustering algorithm $\mathcal{A}$ has a similar feature, i.e., can be viewed as a procedure that first obtains a core of the original graph (by using LP or other methods), and then applies the PIVOT algorithm on the core, then we can get roughly the same approximation ratio as $\mathcal{A}$.

# G ADDITIONAL EXPERIMENTS

In this section, we provide detailed descriptions of the datasets and predictors used in the experiments. Additionally, we present further experimental settings and results.

## G.1 DETAILED DESCRIPTIONS OF DATASETS

In this subsection, we give a detailed description of the real-world datasets used in our experiments. Recall that we use EMAILCORE (Leskovec et al., 2007; Yin et al., 2017), FACEBOOK (McAuley &

Leskovec, 2012), LASTFM (Rozemberczki & Sarkar, 2020), and DBLP (Yang & Leskovec, 2015) from the Stanford Large Network Dataset Collection (Leskovec & Krevl, 2014).

EMAILCORE is a directed network with 1 005 vertices and 25 571 edges. This network is constructed based on email exchange data from a large European research institution. Each vertex represents a person in the institution. There is a directed edge $(u, v)$ in the network if person $u$ has sent at least one email to person $v$.

FACEBOOK is an undirected network with 4 039 vertices and 88 324 edges. This network consists of friend lists of users from Facebook. Each vertex represents a user in Facebook. There is an undirected edge $(u, v)$ in the network if $u$ and $v$ are friends. Due to the computational bottleneck of solving the LP, we only use its three ego-networks: FB 0 ($n = 333, m = 5 038$), FB 414 ($n = 150, m = 3 386$), FB 3980 ($n = 52, m = 292$).

LASTFM is an undirected network with 7 624 vertices and 27 806 edges. This network is a social network of LastFM users, collected from the public API. Each vertex represents a LastFM user from an Asian country. There is an undirected edge $(u, v)$ in the network if $u$ and $v$ are mutual followers.

DBLP is an undirected co-authorship network with 317 080 vertices and 1 049 866 edges. Each vertex represents an author. There is an undirected edge $(u, v)$ in the network if $u$ and $v$ publish at least one paper together. Ground-truth communities are defined based on publication venues: authors who have published in the same journal or conference belong to the same community. For our experiments, we use a sampled subgraph consisting of 2 000 vertices.

**Remark.** We treat the edges in the datasets as positive edges and non-edges as negative implicitly. (For datasets used in experiments where binary classifiers are employed as predictors, the interpretation of positive and negative edges differs slightly. See Appendix G.2 for details.) For directed networks, we convert all directed edges into undirected edges. We highlight that since we are considering labeled complete graphs, the number of edges scales quadratically w.r.t. the number of vertices, which leads to a non-trivial scale of instances.

## G.2 DETAILED DESCRIPTIONS OF PREDICTORS

**Noisy predictor.** We use this predictor for datasets with available optimal clusterings. We form this predictor by performing perturbations on optimal clusterings. Specifically, for any two vertices $u, v \in V$, if $u$ and $v$ are in different clusters in the optimal clustering, then we set the prediction $d_{uv}$ to be $1 - \varepsilon_0$, otherwise $\varepsilon_0$, where $\varepsilon_0 \in (0, 0.5)$. For synthetic datasets with $p = 0.95$, we can assume that the ground truths are also optimal solutions. For real-world datasets, we use the powerful LP solver Gurobi (Gurobi Optimization, LLC, 2023) to get the optimal clusterings.

**Spectral clustering.** We use this predictor for EMAILCORE and LASTFM. It first maps all the vertices to a $d$-dimensional Euclidean space using the graph Laplacian, then clusters all the vertices based on their embeddings. For any two vertices $u, v \in V$, we form the prediction $d_{uv}$ to be $1 - \frac{\langle \boldsymbol{x}_u, \boldsymbol{x}_v \rangle}{\|\boldsymbol{x}_u\| \|\boldsymbol{x}_v\|}$, where $\boldsymbol{x}_u, \boldsymbol{x}_v \in \mathbb{R}^d$ are spectral embeddings of $u$ and $v$, and $\langle \boldsymbol{x}_u, \boldsymbol{x}_v \rangle$ is the dot product of $\boldsymbol{x}_u$ and $\boldsymbol{x}_v$. Note that a larger $d$ indicates a higher-quality predictor.

**Binary classifier.** We use this predictor for datasets where ground-truth communities are available. This predictor is constructed by training a binary classifier (based on an MLP model) to predict whether two vertices belong to the same cluster using node features. In this setting, the goal of Correlation Clustering aligns with that of community detection by treating edges between two vertices in the same (ground-truth) community as positive edges and edges between two vertices in different communities as negative edges. The predictions provided by the binary classifier (i.e., binary values in $\{0, 1\}$) are then used as the pairwise dissimilarities $d_{uv}$ in our algorithms.

## G.3 ADDITIONAL RESULTS

### G.3.1 PERFORMANCE OF ALGORITHM 7 ON REAL-WORLD DATASETS

In this subsection, we present the results of our algorithm in insertion-only streams (Algorithm 7) on real-world datasets, as shown in Figure 3. The results show that under good prediction quality, Algorithm 7 consistently outperforms other baselines across all datasets used. For example, in Figure 3(a), when $\beta \approx 1.2$, the average cost of Algorithm 7 is $13\%$ lower than that of CLMNPT21

and 17% lower than that of CKLPU24. Besides, in Figure 3(c), Algorithm 7 reduces the clustering cost by up to 14% compared to CLMNPT21. Even if the prediction quality is poor, Algorithm 7 does not perform worse than CM23 and achieves comparable performance to CLMNPT21 (on FACEBOOK subgraphs).

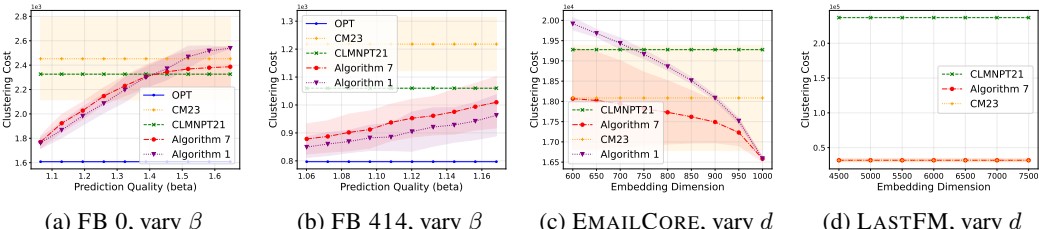

(a) FB 0, vary $\beta$      (b) FB 414, vary $\beta$      (c) EMAILCORE, vary $d$      (d) LASTFM, vary $d$

Figure 3: Performance of Algorithm 7 on real-world datasets. Figures 3(a)–(b) show the effect of prediction quality $\beta$ on two FACEBOOK subgraphs, where we use noisy predictors. Figures 3(c)–(d) examine the effect of the dimension $d$ of spectral embeddings on EMAILCORE and LASTFM, where we use spectral clustering as the predictor. We set $k = 25$ for Figure 3(a), $k = 15$ for Figure 3(b), $k = 10$ for Figure 3(c), and $k = 50$ for Figure 3(d).

### G.3.2 PERFORMANCE OF ALGORITHM 1 ON SYNTHETIC DATASETS WITH VARYING $p$

Recall that in the main text, the experiments on synthetic datasets are conducted only with SBM parameter $p = 0.95$, which is a relatively easy case. In this subsection, we present additional results for smaller values of $p$, as shown in Figure 4. Note that, in these cases, we can no longer assume that the ground truths are also optimal solutions. Therefore, we solve the LP to obtain the optimal solutions, which are required for the noisy predictors. Due to the computational bottleneck of solving the LP, we set $n = 100$. The results demonstrate that even when the ground-truth communities are less obvious (e.g., when $p = 0.7$), the clustering cost of Algorithm 1 is reduced by up to 26% compared to the algorithm of CKLPU24.

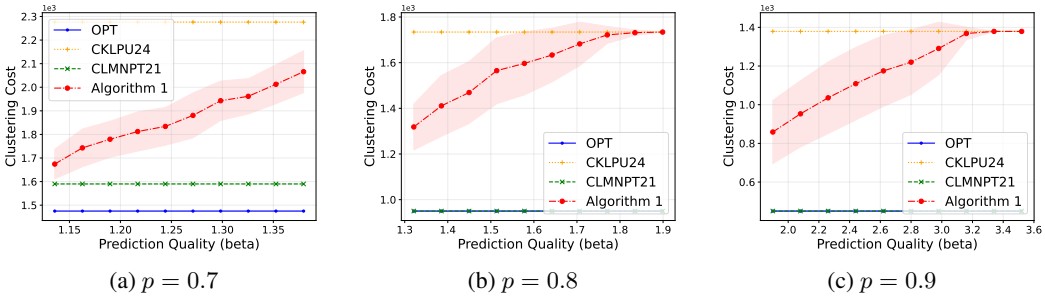

(a) $p = 0.7$             (b) $p = 0.8$             (c) $p = 0.9$

Figure 4: Performance of Algorithm 1 on synthetic datasets with varying values of $p$. We examine the effectiveness of Algorithm 1 when the ground-truth communities are less obvious. We set $n = 100$.

### G.3.3 RUNNING TIME OF OUR ALGORITHMS

In this subsection, we present the running time of our algorithms on FACEBOOK subgraphs, compared to their non-learning counterparts, as shown in Table 2 (Algorithm 1) and Table 3 (Algorithm 7). The results show that our learning-augmented algorithms do not introduce significant time overheads. The slight increase in running time is due to the additional steps of querying the oracles and calculating the costs of two clusterings. These steps are both reasonable and acceptable. Moreover, in the streaming setting, space efficiency is typically the primary focus.

Table 2: Running time (ms) of Algorithm 1 (for dynamic streams) on FACEBOOK subgraphs, compared to its non-learning counterpart. For FB 0, we set $\beta = 1.19$. For FB 414, we set $\beta = 1.12$. For FB 3980, we set $\beta = 1.19$. The reported values are averaged over 20 runs.

| Algorithm     Dataset | FB 0 | FB 414 | FB 3980 |
|---|---|---|---|
| CKLPU24 | 1 738.16 | 165.55 | 7.32 |
| **Algorithm 1** | 1 639.22 | 163.35 | 7.69 |

Table 3: Running time (ms) of Algorithm 7 (for insertion-only streams) on FACEBOOK subgraphs, compared to its non-learning counterpart. For FB 0, we set $\beta = 1.19$. For FB 414, we set $\beta = 1.12$. For FB 3980, we set $\beta = 1.19$. The reported values are averaged over 20 runs.

| Algorithm     Dataset | FB 0 | FB 414 | FB 3980 |
|---|---|---|---|
| CM23 | 30.65 | 6.67 | 0.97 |
| **Algorithm 7** | 81.31 | 16.58 | 2.12 |

### G.3.4 RESULTS BASED ON BINARY CLASSIFICATION PREDICTORS

In this subsection, we present experiments where binary classifiers are employed as predictors in our algorithms. These experiments are performed on three SBM graphs with parameter $p = 0.95$ (each with a different number of vertices) and the DBLP dataset (sampled subgraph of 2 000 vertices). The results are shown in Table 4. The results demonstrate that our learning-augmented algorithms consistently outperform their non-learning counterparts across all datasets. For instance, on the SBM graph with $n = 2400$ vertices, Algorithm 1 reduces the clustering cost by 72% compared to CKLPU24. On the DBLP dataset, Algorithm 7 achieves a 19% reduction in clustering cost compared to CM23.

Table 4: Clustering costs of our algorithms leveraging binary classification predictors, compared to their non-learning counterparts. For Algorithm 7, we set parameter $k = 10$ across all datasets. The reported values are averaged over 5 runs.

| Algorithm     Dataset | SBM $(n = 1200)$ | SBM $(n = 2400)$ | SBM $(n = 3600)$ | DBLP |
|---|---|---|---|---|
| CKLPU24 | 105 269 | 524 800 | 1 114 306 | 7 931 |
| **Algorithm 1** | 35 851 | 145 562 | 324 948 | 7 449 |
| CM23 | 99 273 | 385 736 | 901 631 | 8 452 |
| **Algorithm 7** | 35 851 | 155 335 | 324 948 | 6 862 |

