# OpenReview forum: "Learning-Augmented Streaming Algorithms for Correlation Clustering"
_ICLR.cc/2025/Conference — Submitted to ICLR 2025_

### Official Review · Reviewer_jYz8 · 2024-10-23

**Soundness:** 3
**Presentation:** 3
**Contribution:** 3
**Rating:** 6
**Confidence:** 3

**Summary:**

This paper proposes novel learning-augmented algorithms for correlation clustering, including algorithms in offline as well as dynamic and insert-only streaming settings. All these algorithms are associated with thorough theoretical analyses on approximation factors and complexities. Finally, experiments are conducted to evaluate the performance of the proposed algorithms.

**Strengths:**

S1. As a theoretical paper, this paper proposes novel learning-augmented algorithms with better approximation factors than existing algorithms for the classic correlation clustering problem, showing good originality and technical quality.

S2. This paper is well-written and well-organized.

S3. The theoretical analyses of t are generally sound, though I cannot check every detail of the proofs.

S4. Some experimental results are provided to verify the empirical performance of the proposed algorithms.

**Weaknesses:**

W1. My main concern is that the experimental results provided are quite concise and insufficient.
- The datasets used in the experiments are quite small (with only thousands of vertices). I wonder if streaming algorithms are necessary to process such small graphs since even an $O(|V|^2)$ algorithm can fit in memory.
- Both predictors, namely noisy predictor and spectral clustering, are essentially impractical in (dynamic) streaming. The prior one assumes that the optimal clustering is already available and the latter one should utilize the adjacency matrix as input. From the perspective of online learning, predictions should be made based on prior knowledge about vertices without using global information.
- How about the running time of the proposed algorithms compared to existing ones? Do predictors introduce additional overheads?
- Can the algorithms in offline and insert-only settings serve as baselines? I have had this question since these settings are less general than the dynamic settings. Thus, the proposed algorithms are deemed to be easily adapted to such settings.

W2. Some minor problems with the presentation details.
- The text in the figures is too small and hard to distinguish.
- Add a table to summarize the theoretical results of this paper and compare them with existing ones.

**Questions:**

See the problems in W1.

One additional question:

Q1. Can the proposed algorithms support other types of predictors, such as binary classification?

---

> ### Author Response · Authors · 2024-11-20
> **Official Comment by Authors (1/2)**
>
> We thank the reviewer for the helpful comments. Below please find our responses to the comments.
>
> >[W1] The datasets used in the experiments are quite small (with only thousands of vertices). I wonder if streaming algorithms are necessary to process such small graphs since even an $O(|V|^2)$ algorithm can fit in memory.
>
> We agree that the graphs used in our experiments are relatively small. However, they serve as a useful testbed for evaluating our proposed algorithms. Moreover, since we rely on an LP solver to obtain the optimal solution and generate noisy predictions, the computational bottleneck of solving the LP limits the size of the graphs used in our experiments. Nonetheless, our proposed algorithms are fully capable of processing arbitrarily large graphs in real-world clustering applications.
>
> >[W1] Both predictors, namely noisy predictor and spectral clustering, are essentially impractical in (dynamic) streaming. The prior one assumes that the optimal clustering is already available and the latter one should utilize the adjacency matrix as input. From the perspective of online learning, predictions should be made based on prior knowledge about vertices without using global information.
>
> To clarify, we did not find a suitable dataset that simultaneously provides both pairwise distances and graph structure. Therefore, in our experiments, we used two types of predictors -- noisy predictors based on optimal clustering and spectral clustering based on the adjacency matrix -- purely for evaluating our proposed algorithms. However, our algorithms are designed to work with any predictor that provides pairwise distances, regardless of how it is constructed or obtained.
>
> Furthermore, we note that this paper focuses on the streaming model, which is very different from online learning. In both settings, data is represented as a sequence. In the streaming model, the goal is to scan the sequence in a few number of passes and output the solution at the end of the stream while minimizing space usage. In contrast, online learning makes a prediction at every time step based on the current model (determined by prior data) and then updates the model using the newly observed data.
>
> >[W1] How about the running time of the proposed algorithms compared to existing ones? Do predictors introduce additional overheads?
>
> In the revised version of our paper, we have included results comparing the running time of our algorithms with their non-learning counterparts (see Table 2 and Table 3 in Appendix G.3.3). The results demonstrate that our learning-augmented algorithms do not introduce significant time overheads. The slight increase in running time is due to the additional steps of querying the oracles and calculating the costs of two clusterings. These steps are both reasonable and acceptable. Moreover, in the streaming setting, space efficiency is typically the primary focus.
>
> >[W1] Can the algorithms in offline and insert-only settings serve as baselines? I have had this question since these settings are less general than the dynamic settings. Thus, the proposed algorithms are deemed to be easily adapted to such settings.
>
> In the revised version of our paper, we have included the results of Algorithm 1 (designed for dynamic streams) as baselines in Figures 1(c)(d) and Figures 3(a)(b)(c). The results show that when applied to insertion-only streams, Algorithm 1 performs worse than Algorithm 7 (designed for insertion-only streams) in some cases. This is likely because Algorithm 1 includes certain components necessary for handling edge deletions in dynamic streams, which are unnecessary in insertion-only streams and may incur additional cost. Additionally, we remark that our streaming algorithms are direct implementations of their offline counterparts, with only minor modifications to accommodate the streaming model. The approximation ratios remain identical. Since we focus on streaming algorithms, we have chosen not to include offline algorithms as baselines to ensure a fair comparison within the streaming model.
>
> >[W2] Minor: The text in the figures is too small and hard to distinguish. Add a table to summarize the theoretical results of this paper and compare them with existing ones.
>
> Thank you for the valuable suggestions! We have redrawn all the figures and updated them in the revised version. Additionally, in the Introduction section, we have added a table that compares the best-known results (without predictions) with ours.

---

> ### Author Response · Authors · 2024-11-20
> **Official Comment by Authors (2/2)**
>
> >[Q1] Can the proposed algorithms support other types of predictors, such as binary classification?
>
> We are not entirely sure about the meaning of binary classification here, as for correlation clustering, the most natural case is when the number of clusters is not specified. Thus, a predictor that only predicts if a vertex belongs to the 1st cluster or the 2nd cluster seems not very helpful. On the other hand, if binary classification refers to predicting, for each edge, whether its two endpoints belong to the same cluster (i.e., a binary value in {$0, 1$}), then our algorithm can utilize such information, provided that the predictions satisfy the triangle inequality.

---

> > ### Comment · Reviewer_jYz8 · 2024-11-21
> >
> > When there exist ground-truth communities in the graph, a binary classification model can be trained to predict whether two nodes belong to the same community using node features. More generally, a classification model can predict the community label of a node based on its features. In such cases, the goal of correlation clustering can be aligned with the goal of community detection by considering edges between two nodes in the same community as "positive edges" and edges between two nodes in different communities as "negative edges". I think performing experiments in such settings is useful to evaluate the performance of the algorithm in a more realistic setting following a common concept of "predictor".

---

> > > ### Author Response · Authors · 2024-11-25
> > >
> > > Thank you for your quick response and clarification. Following your suggestion, we have conducted additional experiments where a binary classifier (based on an MLP model) is trained to predict whether two vertices belong to the same cluster using node features. Specifically, as you mentioned, we treat edges between two nodes in the same (ground-truth) community as "positive edges" and edges between two nodes in different communities as "negative edges". The predictions made by the binary classifier (i.e., binary values in {$0, 1$}) are then used as the pairwise dissimilarities $d_{uv}$ in our algorithms.
> > >
> > > These experiments were conducted on three SBM graphs with parameter $p=0.95$ (each with a different number of vertices) and the DBLP dataset (sampled subgraph of $2000$ vertices). The clustering costs of our algorithms compared to their non-learning counterparts are as follows:
> > >
> > > - SBM ($n=1200$):
> > >   - CKLPU24: 105269
> > >   - Algorithm 1 (Our algorithm in dynamic streams): 35851
> > >   - CM23: 99273
> > >   - Algorithm 7 (Our algorithm in insertion-only streams): 35851
> > >
> > > - SBM ($n=2400$):
> > >   - CKLPU24: 524800
> > >   - Algorithm 1 (Our algorithm in dynamic streams): 145562
> > >   - CM23: 385736
> > >   - Algorithm 7 (Our algorithm in insertion-only streams): 155335
> > >
> > > - SBM ($n=3600$):
> > >   - CKLPU24: 1114306
> > >   - Algorithm 1 (Our algorithm in dynamic streams): 324948
> > >   - CM23: 901631
> > >   - Algorithm 7 (Our algorithm in insertion-only streams): 324948
> > >
> > > - DBLP dataset:
> > >   - CKLPU24: 7931
> > >   - Algorithm 1 (Our algorithm in dynamic streams): 7449
> > >   - CM23: 8452
> > >   - Algorithm 7 (Our algorithm in insertion-only streams): 6862
> > >
> > > The results demonstrate that our learning-augmented algorithms consistently outperform their non-learning counterparts across all datasets. For instance, on the SBM graph with $n=2400$ vertices, Algorithm 1 reduces the clustering cost by 72% compared to CKLPU24. On the DBLP dataset, Algorithm 7 achieves a 19% reduction in clustering cost compared to CM23.
> > >
> > > We have included this setting (see Appendix G.2) and the corresponding results (see Appendix G.3.4) in the revised version of our paper.

---

> > > > ### Comment · Reviewer_jYz8 · 2024-11-27
> > > >
> > > > Thanks for the rebuttal. My concerns about experiments are mostly addressed.

---

### Official Review · Reviewer_1U7n · 2024-11-01

**Soundness:** 3
**Presentation:** 3
**Contribution:** 3
**Rating:** 6
**Confidence:** 4

**Summary:**

The paper studied learning-augmented streaming correlation clustering. In this problem, a labeled complete graph $G=(V, E^+ \cup E^-)$ is given as a stream of edges (together with the labels at each point), and the goal is to minimize the *disagreement cost*, defined as the total number of $(+)$ edges crossing clusters and the number of $(-)$ edges in the same clusters are minimized. For the streaming setting, there exists an algorithm by CLPTYZ [STOC’24] that achieves $(2-2/13+\varepsilon)$ approximation with O(n log n) space (note that although this is claimed to be the sate-of-the-art, there actually exists an algorithm that gives $(1+\varepsilon)$-approximation with $\tilde{O}(n)$ space, albeit a very impractical one. See the ‘questions’ section for more on this).

The algorithm in CLPTYZ [STOC’24] is fairly involved and hides a large constant. As such, motivated by the practical aspect of the problem, the paper considered correlation clustering with learning-augmented oracles. Specifically, it introduces a similarity oracle that, when queried for a vertex pair $(u,v)$, provides a rough estimate $d_{u,v}$, representing the "distance" between $u$ and $v$. The quality of the oracle is measured by the ‘$\beta$ level’, which is defined by an upper bound of $\beta\cdot \text{OPT}$ on total $d_{u,v}$​ values on $(+)$ edges and $(1 - d_{u,v})$ values on $(-)$ edges. Using such a predictor, the algorithm that achieves a $(\min\{2.06\beta, 3\}+\varepsilon)$-approximation in $O(n \log n)$ space. The space complexity is considered much more practical than the CLPTYZ [STOC’24] algorithm.

The paper further conducted experiments on both synthetic and real-world graphs. The performance is better than pivot-based algorithms and is comparable to the agreement decomposition-based algorithm of CLMNPT [ICML’21].

**Strengths:**

Learning-augmented correlation clustering is a natural problem, and I believe people have explored different notions of predictions to find the most ‘natural’ oracle to be used (and it has to be practical). This paper made a concrete step in the right direction. The oracle is sufficiently natural, although the quality notion of $\beta$-level is a bit artificial. The experiments have shown that the oracle could help improve the performances of pivot-based algorithms significantly. As such, I think the paper could be a good addition to the literature.

**Weaknesses:**

I do not believe I could strongly support the paper due to the following reasons.
- Novelty: although the paper claims to be technically novel and the analysis is very non-trivial, I believe the novelty of the algorithm is quite limited. The algorithm is essentially formed by the truncation during the stream and the post-processing part. For the streaming phase, the idea is mainly from CKLPU [SODA’24] (cf. CM [NeurIPS’23]), and I do not see anything that is substantially novel. The baseline (no worse than $3$ approximation) basically comes for free by running the algorithm of CKLPU [SODA’24], and the space analysis simply follows. The post-processing algorithm adapts the algorithm of CMSY [STOC’15], for which I will give some credit for novelty. However, the key insights for this part were not clearly articulated in the paper, and I had to go to page 18 in the appendix to see why the approximation guarantee holds.
- The $2.06\beta$ approximation factor is also somehow weak: to outperform the pivot-based $3$-approximation, there has to be $\beta \leq \frac{3}{2.06}<1.5$, which means you need predictions with very high quality. This is also clear from your experiments that your algorithm only outperforms the agreement-based algorithm when $\beta$ is quite low.
- I also have a criticism for the learning-augmented framework being used on *graph streaming* algorithms. The streaming model mainly aims to optimize space efficiency; however, for graph-related applications, if you train a model using ML, it’s most likely some variates of graph neural networks. To train these models you would naturally need to preserve almost all edges, which is a great cost in terms of space. I think for the oracle proposed in this paper, maybe you could train some ML model that is more ‘local’ and only answers pairs of vertices. However, in such a case, how do you ensure the triangle inequality? Anyway, I think some aspects related to this point should be discussed in the paper.

**Questions:**

- Do you have any responses to the weakness (especially the question about how to efficiently implement the oracle)?
- (A comment:) Note that if we just aim for low-space single-pass streaming algorithm correlation clustering, there is a deterministic algorithm in "Single-Pass Streaming Algorithms for Correlation Clustering" (BCMT [SODA’23]) that returns a $(1+\varepsilon)$-approximation, albeit with very terrible ($n^n$) time. The algorithm simply enumerates all possible clusterings and checks their costs to output the min one with the cut sparsifier. This algorithm has no way to be implemented in practice, but it is indeed unbeatable in terms of space complexity and approximation. This is related to the ‘practical’ aspects of your paper, and maybe you want to discuss that also in your paper.

---

> ### Author Response · Authors · 2024-11-20
>
> We thank the reviewer for the helpful comments. Below please find our responses to the comments.
>
> >The post-processing algorithm adapts the algorithm of CMSY [STOC’15], for which I will give some credit for novelty. However, the key insights for this part were not clearly articulated in the paper, and I had to go to page 18 in the appendix to see why the approximation guarantee holds.
>
> Thank you for the suggestion! In the revised version, we have moved the proof of Lemma 3.6 from the appendix to the main text, which clearly explains the key insights behind why the approximation guarantee holds.
>
> >I also have a criticism for the learning-augmented framework being used on graph streaming algorithms. The streaming model mainly aims to optimize space efficiency; however, for graph-related applications, if you train a model using ML, it’s most likely some variates of graph neural networks. To train these models you would naturally need to preserve almost all edges, which is a great cost in terms of space. I think for the oracle proposed in this paper, maybe you could train some ML model that is more ‘local’ and only answers pairs of vertices. However, in such a case, how do you ensure the triangle inequality? Anyway, I think some aspects related to this point should be discussed in the paper.
>
> Thank you for the valuable comment! As we motivated in the Introduction, to cluster a graph $G$, we may use ML methods to train some other related networks that are defined on the same vertex set $V(G)$, to learn the pairwise (dis)similarities $d_{uv}$. In particular, we can learn the node embeddings from these related networks, which map all vertices to Euclidean space. Then the distances $d_{uv}$ between these points serve naturally as pairwise dissimilarities and satisfy the triangle inequality.
>
> Furthermore, we note that as in many other learning augmented streaming algorithms [1,2,3,4], we do not count the space of the oracle or the space of the algorithm for constructing such an oracle in our streaming algorithms.
>
> [1] Chen-Yu Hsu, Piotr Indyk, Dina Katabi, and Ali Vakilian. Learning-Based Frequency Estimation Algorithms. ICLR 2019.
>
> [2] Tanqiu Jiang, Yi Li, Honghao Lin, Yisong Ruan, and David P. Woodruff. Learning-Augmented Data Stream Algorithms. ICLR 2020.
>
> [3] Justin Y. Chen, Talya Eden, Piotr Indyk, Honghao Lin, Shyam Narayanan, Ronitt Rubinfeld, Sandeep Silwal, Tal Wagner, David P. Woodruff, and Michael Zhang. Triangle and Four Cycle Counting with Predictions in Graph Streams. ICLR 2022.
>
> [4] Anders Aamand, Justin Y. Chen, Huy Lê Nguyen, Sandeep Silwal, and Ali Vakilian. Improved Frequency Estimation Algorithms with and without Predictions. NeurIPS 2023.
>
> >Note that if we just aim for low-space single-pass streaming algorithm correlation clustering, there is a deterministic algorithm in "Single-Pass Streaming Algorithms for Correlation Clustering" (BCMT [SODA’23]) that returns a $(1+\varepsilon)$-approximation, albeit with very terrible ($n^n$) time. The algorithm simply enumerates all possible clusterings and checks their costs to output the min one with the cut sparsifier. This algorithm has no way to be implemented in practice, but it is indeed unbeatable in terms of space complexity and approximation. This is related to the ‘practical’ aspects of your paper, and maybe you want to discuss that also in your paper.
>
> Thank you for pointing this out; it helps us better motivate the problem and our results. We have added the discussion in the revised version of the paper.

---

> > ### Comment · Reviewer_1U7n · 2024-11-21
> >
> > Thank you for the updates and rebuttal. I want to point out that all the references you listed are learning-augmented algorithms for *itemized* data instead of graph data. For graph-based data, there is an actual concern that the model might have to work with $\Theta(n^2)$ edges. For your application, I agree that maybe you could use embeddings on the nodes to obtain the predictions you are looking for, but I'm not sure how accurate it will be.
> >
> > I also appreciate the authors' update on the discussion about Lemma 3.6 and the $(1+\varepsilon)$-approximation. I do not think anything substantial has been changed, though.
> >
> > With all the pros and cons being stated, I'll keep my evaluation as it is.

---

### Official Review · Reviewer_e84w · 2024-11-02

**Soundness:** 4
**Presentation:** 2
**Contribution:** 2
**Rating:** 6
**Confidence:** 3

**Summary:**

The paper presents new algorithms for correlation clustering in dynamic graph streams where both edge insertions and deletions are allowed. The algorithms leverage prediction models for the pairwise similarity between nodes. It is shown that under certain assumptions about the properties of the predictor, the algorithm can achieve better-than-3 approximation of the optimal correlation clustering cost. The main contribution is rigorous theoretical results for the approximation quality of the presented algorithms. Experimental results support the theoretical findings.

**Strengths:**

- The paper studies an important problem
- An interesting paper exploring a promising direction, namely how to leverage prediction models for combinatorial hard optimization problems.
- Rigorous theoretical results.
- The algorithms are easy to implement yet the analysis uses advanced tools to derive the aforementioned theoretical results.

**Weaknesses:**

I have one main problem with the paper. Namely, the definition of a $\beta$-level predictor looks artificial and does not reflect some natural properties of the data. With this definition, the result essentially becomes something like "If we can predict similarities that could be used to obtain a good approximation of the optimal solution, then we can obtain a good approximation algorithm". Of course, I am oversimplifying but I think the authors should try to provide some connection between the formal definition and some intuitive properties of the data and the prediction algorithms. I realize this is challenging but even a special example would be helpful.

Furthermore, the pairwise similarities $d_{uv}$ should be thoroughly discussed and concrete examples of similarity functions should be presented. More specifically, the relation between the edge labels and $d_{uv}$ similarities should be discussed with concrete examples.

Some minor comments:
- The optimization objective is never formally defined.
- Algorithm 1 assumes that the number of nodes $n$ is known in advance. I guess this is not a real limitation but might add another logarithmic factor to the time and space complexity. Please comment on it.

**Questions:**

Comment on the above weaknesses. In particular, it would be great if you could provide some intuition with concrete examples about $d_{uv}$ and the optimal clustering.

---

> ### Author Response · Authors · 2024-11-20
>
> We thank the reviewer for the helpful comments. Below please find our responses to the comments.
>
> >Discussion of the $\beta$-level predictor & pairwise dissimilarities $d_{uv}$.
>
> As stated in the Introduction, it is quite common that multiple networks are defined over the same set of vertices. It is natural to leverage machine learning techniques to learn the pairwise (dis)similarities between vertices using one or more of these networks. Leveraging these similarities across networks can greatly aid in exploring the cluster structure of any newly defined network over the same set of vertices. For example, one may have applied some node embedding techniques, i.e., mapping each vertex to some point in Euclidean space. This embedding naturally gives pairwise distance $d_{uv}$ of the points corresponding to $u$ and $v$, which can be viewed as a concrete example of dissimilarity functions.
>
> Therefore, the (dis)similarities describe the general relationship (not limited to the input graph) between vertices, whereas the edge labels in the input graph represent the specific relationships relevant to the clustering problem. Our algorithms use both information to help cluster the input graph more effectively.
>
> Regarding the intuition of the $\beta$-level predictor, as stated after Definition 2.1, a smaller $\beta$ intuitively indicates a higher-quality predictor. In this case, $d_{uv}$ captures how likely $u$ and $v$ belong to the same cluster in the optimal solution. This aligns with the intuition in many works on learning-augmented algorithms [1,2], which assume that predictions are small perturbations of the optimal solution.
>
> [1] Jon C. Ergun, Zhili Feng, Sandeep Silwal, David P. Woodruff, and Samson Zhou. Learning-Augmented $k$-means Clustering. ICLR 2022.
>
> [2] Yuko Kuroki, Atsushi Miyauchi, Francesco Bonchi, and Wei Chen. Query-Efficient Correlation Clustering with Noisy Oracle. NeurIPS 2024.
>
> >Minor: The optimization objective is never formally defined.
>
> Thank you for the suggestion! In the revised version, we have added the optimization objective in the first paragraph of the Introduction section.
>
> >Minor: Algorithm 1 assumes that the number of nodes $n$ is known in advance. I guess this is not a real limitation but might add another logarithmic factor to the time and space complexity. Please comment on it.
>
> We note that it suffices to know an upper bound (a constant-factor approximation) of $n$. Otherwise, it is unclear whether the current algorithm can be modified to work, as it requires picking a random permutation of all the vertices at the beginning.

---

> > ### Comment · Reviewer_e84w · 2024-11-24
> >
> > I thank the authors for their responses. I want to keep my score.  I understand that defining predictors as some kind of noisy oracles has been used in previous works but this doesn't address my concern. For example, if you show that for certain graph classes with real-life properties, e.g. high clustering coefficient, the spectral clustering predictor satisfies the formal definition of a $\beta$-level predictor, this would make the results much more convincing.

---

> > > ### Author Response · Authors · 2024-11-25
> > >
> > > Thank you for your thoughtful feedback. For a class of graphs known as well-clustered graphs [3], the predictions provided by spectral clustering are likely to exhibit good quality. Broadly speaking, well-clustered graphs are characterized by strong clusterability, where edges are dense within the same latent cluster and sparse between different latent clusters. (In the context of Correlation Clustering, we implicitly treat the edges in the graph as positive edges and non-edges as negative edges.)
> > >
> > > Recall that the spectral clustering predictor maps all vertices to a $d$-dimensional Euclidean space via the graph Laplacian and then clusters them based on their embeddings. Theoretically, it can be shown that vertices within the same latent cluster are embedded into points that are close to each other, while vertices from different latent clusters are embedded into points that are far apart. For any two vertices $u,v\in V$, the prediction $d_{uv}$ is given by $d_{uv}=1-\frac{\left\langle x_u, x_v\right\rangle}{||x_u|| \cdot ||x_v||}$, where $x_u, x_v \in \mathbb{R}^d$ are spectral embeddings of $u$ and $v$. As a result, the total sum $\sum_{(u, v) \in E^{+}} d_{u v}+\sum_{(u, v) \in E^{-}}(1-d_{u v})$ is small on well-clustered graphs, suggesting that the spectral clustering predictor is a reasonable choice for these graphs.
> > >
> > > We believe this example highlights that our results are applicable to certain types of graphs with real-world properties. We hope this, along with previous responses, could address your concerns.
> > >
> > > [3] Richard Peng, He Sun, and Luca Zanetti. Partitioning Well-Clustered Graphs: Spectral Clustering Works! COLT 2015 & SIAM Journal on Computing 2017.

---

### Official Review · Reviewer_z4se · 2024-11-03

**Soundness:** 4
**Presentation:** 3
**Contribution:** 3
**Rating:** 6
**Confidence:** 3

**Summary:**

The authors propose a learning augmented algorithm (LAG) for correlation clustering in the streaming setting. More specifically, the authors propose a min{(2.06\beta, 3)} + \eps algorithm for the problem. Further, they propose a sparsification scheme to approximate the clustering in the original graph to within (1 \pm \eps) to deal with memory constraints in the streaming setting.

**Strengths:**

The authors adapt two algorithms from prior work and extend it to the case where the algorithm has access to predictions on the pairwise dissimilarity between nodes in the graph.  They present a clean algorithm that clearly outperforms the best-known algorithm when the predictions are good.  This is an important use case of the LAG framework.  In fact, I am surprised that this was not studied earlier in the LAG framework.

The paper is well-written with corroborating empirical analysis that shows the efficiency of the proposed algorithms.

**Weaknesses:**

The streaming setting and the use of graph sparsification is a bit weak, IMO.  By sparsification, I would think they have completely avoided the subtle issues that arise from the streaming setting (e.g., order of arrival of the edges, etc).

**Questions:**

How does one go back to the original clustering from the sparse clustering computed by the algorithm?

---

> ### Author Response · Authors · 2024-11-20
>
> >The streaming setting and the use of graph sparsification is a bit weak, IMO. By sparsification, I would think they have completely avoided the subtle issues that arise from the streaming setting (e.g., order of arrival of the edges, etc).
>
> Thank you for your comment. While we are not entirely sure we fully understand it, we want to clarify that, like other streaming algorithms, we do need to account for the order of arrival of the edges. Could you please clarify your question in a bit more detail? This would help us address your concerns more effectively.
>
> >How does one go back to the original clustering from the sparse clustering computed by the algorithm?
>
> We note that the sparsification is applied to the edges rather than the vertices. The clustering computed on the sparsified graph is a partition of the vertex set $V$, which directly corresponds to the clustering of the original graph.

---

> > ### Comment · Reviewer_z4se · 2024-11-27
> >
> > Thanks for the clarification.   Some details on how an effective predictor can be practically implemented will help.

---

> > > ### Author Response · Authors · 2024-11-27
> > >
> > > Thank you for your response! As we motivated in the Introduction, to cluster a graph $G$, we may use machine learning techniques to train some other related networks that are defined on the same vertex set $V(G)$, to learn the pairwise (dis)similarities $d_{uv}$. In particular, we can learn the node embeddings from these related networks, which map all vertices to Euclidean space. Then the distances $d_{uv}$ between these points serve naturally as pairwise dissimilarities and satisfy the triangle inequality.

---

### Official Review · Reviewer_KzdP · 2024-11-05

**Soundness:** 3
**Presentation:** 2
**Contribution:** 3
**Rating:** 6
**Confidence:** 3

**Summary:**

The paper studies the correlation clustering problem, proposing a learning-augmented streaming algorithm for the problem. The proposed algorithm can achieve better-than 3-approximation in dynamic settings, and the structure of the proposed algorithm is simpler than previous algorithm. Experimental results are also presented in the paper to empirically testify the performance of the proposed algorithm.

**Strengths:**

S1. Correlation clustering is a fundamental and widely studied problem. The paper is well-motivated to study this problem and simplify previous algorithms.

S2. The proposed algorithm can achieve better approximation results when the predictor works well. The empirical experiments also show the effectiveness of the proposed algorithm.

**Weaknesses:**

W1. I have concerns about the novelty and technical depth of the proposed algorithms. The core idea is largely based on existing approaches, and the main theoretical results are derived from previous work, which limits the paper’s novelty. Additionally, the theoretical improvement achieved by this work appears to be marginal; the proposed algorithm performs asymptotically better than previous results only when the predictor functions well.

W2. The structure of the proposed algorithm is simpler than previous approaches, which is good. However, this simplification results in a loss of approximation quality (i.e., it does not achieve the 1.827-approximation), which weakens the paper’s contributions.

W3. The paper’s presentation needs significant improvement. I recommend starting with a formal problem definition. Illustrative figures would also be helpful for explaining the correlation clustering problem to non-expert readers. The detailed review of related work in the Introduction could be moved to a separate Related Work section, leaving only a complexity comparison table in the Introduction to emphasize the technical improvements of this manuscript. Presenting high-level ideas first, followed by detailed technical proofs, would also improve readability.

-----
The authors’ rebuttal partially addresses my concerns, and I have accordingly updated my score.

**Questions:**

Could the authors further clarify the technical improvements of the proposed algorithms compared to the recent 1.847-approximation streaming algorithm by Cohen-Addad et al. (STOC ’24)?

---

> ### Author Response · Authors · 2024-11-20
>
> We thank the reviewer for the helpful comments. Below please find our responses to the comments.
>
> >[W1] The theoretical improvement achieved by this work appears to be marginal; the proposed algorithm performs asymptotically better than previous results only when the predictor functions well.
>
> We highlight that consistency and robustness are key features desired in learning-augmented algorithms. Specifically, such algorithms can outperform the best-known results under good prediction quality; even if the prediction quality is poor, they still retain strong worst-case guarantees. Therefore, it is natural that our learning-augmented algorithm is better than previous results when the predictor functions well; if the predictor is bad, then it does not help.
>
> >[W2] The structure of the proposed algorithm is simpler than previous approaches, which is good. However, this simplification results in a loss of approximation quality (i.e., it does not achieve the 1.847-approximation), which weakens the paper’s contributions.
>
> Indeed, our algorithm in insertion-only streams does not achieve the $1.847$-approximation (Theorem 1.2). However, it is important to note that the $1.847$-approximation algorithm cannot be applied to dynamic streams. The best-known approximation ratio for dynamic streams is $(3+\varepsilon)$ by Cambus et al. (SODA’24). Our algorithm in dynamic streams is the first to achieve a better-than-$3$-approximation (Theorem 1.1).
>
> >[W3] The paper’s presentation needs significant improvement. I recommend starting with a formal problem definition. Illustrative figures would also be helpful for explaining the correlation clustering problem to non-expert readers. The detailed review of related work in the Introduction could be moved to a separate Related Work section, leaving only a complexity comparison table in the Introduction to emphasize the technical improvements of this manuscript. Presenting high-level ideas first, followed by detailed technical proofs, would also improve readability.
>
> Thank you for the valuable suggestions! In the Introduction section of the revised version, we have added a formal problem definition and a table that compares the best-known results (without predictions) with ours. Additionally, we have enriched the content of the Other Related Work section in Appendix A.
>
> >Could the authors further clarify the technical improvements of the proposed algorithms compared to the recent 1.847-approximation streaming algorithm by Cohen-Addad et al. (STOC ’24)?
>
> We emphasize that our algorithm is technically incomparable with the $1.847$-approximation streaming algorithm by Cohen-Addad et al. (STOC’24). Specifically, our technical improvements build upon the work of Cambus et al. (SODA’24) and Chawla et al. (STOC’15), leveraging the truncated Pivot algorithm. However, the $1.847$-approximation algorithm is based on local search, which is fundamentally different from ours and is restricted to insertion-only streams. In particular, their algorithm requires enumerating a large number of subsets of a constant-size set $S$, which is considered to be quite impractical and hard for implementation. In contrast, our algorithm is much simpler and can work in dynamic streams.

---

> > ### Comment · Reviewer_KzdP · 2024-11-24
> > **Response to the Rebuttal**
> >
> > Thank you for the rebuttal!
> >
> > The response partially addresses my questions. While my concerns regarding W1 still remain, I am inclined to slightly increase my score.

---

### Author Response · Authors · 2024-11-20

Dear Reviewers,

We would like to express our sincere gratitude for your valuable feedback on our paper. We have provided detailed responses to each of your comments and have uploaded a revised version of the paper, with the changes highlighted in blue.

---

### Meta-Review · Area_Chair_LDjd · 2024-12-17

**Metareview:**

This paper studies the correlation clustering problem in the streaming setting and proposes a learning-augmented algorithm (LAG) to address it. The authors claim their algorithm achieves a better-than-3 approximation in dynamic settings with a simpler structure than previous approaches. They provide experimental results to support their claims.

Reviewers acknowledge that correlation clustering is a fundamental and well-studied problem and appreciate the sound theoretical analysis.

However, they also raise several concerns:

- Limited Practical Relevance: The focus on correlation clustering on cliques is primarily a theoretical problem. Existing streaming algorithms already offer better guarantees for this specific case, raising questions about the practical significance of this work.
- Limited Novelty: The novelty is somewhat limited, as the algorithm builds upon existing techniques and incorporates a learning component in a relatively straightforward manner.
- Unnatural ML Oracle: The ML oracle used by the algorithm seems somewhat unnatural and its practical implementation may be challenging.

Recommendation:

While the paper presents a learning-augmented algorithm for correlation clustering with some theoretical guarantees, the reviewers do not find it strong enough for acceptance at ICLR.

**Additional Comments On Reviewer Discussion:**

The paper was discussed at length and the reviewers agree on the final decision.

---

### Decision · Program_Chairs · 2025-01-22

Reject